# BiVO_4_ As a Sustainable and Emerging Photocatalyst: Synthesis Methodologies, Engineering Properties, and Its Volatile Organic Compounds Degradation Efficiency

**DOI:** 10.3390/nano13091528

**Published:** 2023-05-01

**Authors:** Ganesh S. Kamble, Thillai Sivakumar Natarajan, Santosh S. Patil, Molly Thomas, Rajvardhan K. Chougale, Prashant D. Sanadi, Umesh S. Siddharth, Yong-Chein Ling

**Affiliations:** 1Department of Engineering Chemistry, Kolhapur Institute of Technology’s College of Engineering (Autonomous), Kolhapur Affiliated Shivaji University Kolhapur Maharashtra, Kolhapur 416004, Maharashtra, India; 2Environmental Science Laboratory, CSIR-Central Leather Research Institute (CSIR-CLRI), Chennai 600020, Tamil Nadu, India; 3Academy of Scientific and Innovative Research (AcSIR), Ghaziabad 600113, Uttar Pradesh, India; 4Department of Applied Mechanics, ECTO Group, FEMTO-ST Institute, 24, Rue de l’Epitaph, 25000 Besançon, France; 5School of Studies in Chemistry & Research Centre, Maharaja Chhatrasal Bundelkhand University, Chhatarpur 471001, Madhya Pradesh, India; 6Department of Basic Sciences and Humanities, Sharad Institute of Technology College of Engineering Yadrav (Ichalkaranji), Ichalkaranji 416115, Maharashtra, India; 7Department of Chemistry, National Tsing Hua University, Hsinchu 300044, Taiwan

**Keywords:** BiVO_4_, advanced oxidation process (AOPs), photocatalysis, volatile organic compounds (VOCs), degradation

## Abstract

Bismuth vanadate (BiVO_4_) is one of the best bismuth-based semiconducting materials because of its narrow band gap energy, good visible light absorption, unique physical and chemical characteristics, and non-toxic nature. In addition, BiVO_4_ with different morphologies has been synthesized and exhibited excellent visible light photocatalytic efficiency in the degradation of various organic pollutants, including volatile organic compounds (VOCs). Nevertheless, the commercial scale utilization of BiVO_4_ is significantly limited because of the poor separation (faster recombination rate) and transport ability of photogenerated electron–hole pairs. So, engineering/modifications of BiVO_4_ materials are performed to enhance their structural, electronic, and morphological properties. Thus, this review article aims to provide a critical overview of advanced oxidation processes (AOPs), various semiconducting nanomaterials, BiVO_4_ synthesis methodologies, engineering of BiVO_4_ properties through making binary and ternary nanocomposites, and coupling with metals/non-metals and metal nanoparticles and the development of Z-scheme type nanocomposites, etc., and their visible light photocatalytic efficiency in VOCs degradation. In addition, future challenges and the way forward for improving the commercial-scale application of BiVO_4_-based semiconducting nanomaterials are also discussed. Thus, we hope that this review is a valuable resource for designing BiVO_4_-based nanocomposites with superior visible-light-driven photocatalytic efficiency in VOCs degradation.

## 1. Introduction

In the 21st century, environmental protection and remediation are the greatest challenges for human beings due to massive population increases and the growth of industrialization [1]. Our natural water has been significantly damaged and continues to deteriorate due to human activity and the growth of chemical, agriculture, and pharmaceutical industries. Textile industries, in particular, discharge annually 15% (one thousand tons) of hazardous dyes as effluents [2]. Along with these, a vast amount of toxic and harmful chemicals, heavy metals [3], ionic liquids [4], surfactants [5] agrochemicals, and pharmaceutical industry wastes, including drugs and antibiotics wastes [6], are discharged into fresh water. Organic effluents are highly carcinogenic, leading to a negative impact on the sustainability of water resources [7]. It is anticipated that up to 50% of people will face clean water disasters by 2025 because approximately 70% of industrial wastewater is not well treated and waste effluent is released directly into freshwater sources, causing the severe pollution of natural water bodies [8]. Therefore, the removal of these hazardous pollutants from wastewater before their discharge into the environment is a remarkable challenge across the globe to protect the environment and natural water resources.

Volatile organic compounds (VOCs) are one of the most hazardous organic effluents discharged directly into water by the paint, pharmaceutical, chemical, mining, printing, and petrochemical industries. The most common VOCs used industrially are acetone, formaldehyde, benzene, toluene, xylene, 1,3-butadiene, ethylene glycol, perchloroethylene, methylene chloride, and chlorobenzene, which cause severe health hazards to human health, including toxic, carcinogenic, mutagenic, and teratogenic effects [9,10]. For the elimination of VOCs, over the years, various environmental pollutant remediation technologies, such as membrane separation, adsorption, biological oxidation, and chemical oxidation, have been used, but these can generate secondary pollutants, which require post-treatment and are inefficient for field-level application [11,12]. In order to overcome these problems, heterogeneous semiconductor photocatalysis is one of the promising advanced oxidation processes (AOPs) used for the removal of VOCs and pollutants from wastewater that has been studied as it has the advantages of an environmentally friendly process, low cost, mild reaction conditions, no generation of secondary pollution, and greater effectiveness in the degradation of even low concentrations of pollutants compared to conventional technologies [13,14,15,16,17]. Various well-known stable metal oxides (e.g., TiO_2_, WO_3_, and ZnO, etc.), metal titanates (SrTiO_3_), etc., have been developed and have demonstrated an efficient photocatalytic performance for the degradation of various organic compounds. However, the photocatalytic efficiency of these metal oxides is limited due to their large band gap energy (3.2 eV), response to ultraviolet (UV) light only, and utilization of less than 4% of the available solar spectrum, which restricts their potential commercial application [18,19,20,21]. Therefore, during the last decade, research has been focused on constructing various visible-light-responsive semiconductor photocatalysts, such as metal sulfides, oxides, oxynitrides, chalcogenides, halides, and oxyhalides, for the degradation of hazardous organic pollutants [22,23,24,25,26,27,28,29,30]. However, metal sulfides and some chalcogenides-based catalytic systems show poor stability under light irradiation, which restricts their commercial feasibility. Therefore, the development of metal-oxide-based, visible-light-responsive photocatalysts, especially bismuth-based metal oxide (BiVO_4_, Bi_2_MoO_6_, Bi_2_WO_6_, BiFe_2_O_3_, BiFe_4_O_9_, and BiOX (X = Cl, Br, I, etc.,)) systems, has received significant attention, and these are being used for the degradation of various organic pollutants, including VOCs [31,32,33,34,35,36].

Among bismuth metal oxides, bismuth vanadate (BiVO_4_) has attracted significant interest due to its outstanding visible light absorption (~2.3–2.4 eV band gap energy), suitable band potentials, facet-dependent catalytic activity, non-toxicity, and resistance to photo and chemical corrosion [37,38,39,40,41,42]. Kudo et al., in 1999, developed highly crystalline monoclinic and tetragonal BiVO_4_ by changing the ratio of vanadium to bismuth in starting materials, producing O_2_ evolution under visible light irradiation. Subsequently, significant research efforts have been made to develop BiVO_4_-based systems for the degradation of pollutants [43]. However, there is a high recombination rate of photo-generated charge carriers, and the conduction band position of BiVO_4_ is lower than the superoxide radical anion production potential, which decreases the photocatalytic activity. Consequently, different modifications to BiVO_4_, such as metal and non-metal loading, the control of morphologies, and the formation of heterojunctions with bismuth and non-bismuth-based oxides, have been performed, which has led to various reviews and research articles on the degradation of water pollutants. However, review articles on BiVO_4_-based materials for VOCs degradation are rarely reported. Therefore, the present review covers recent developments in BiVO_4_-based materials and their potential visible light photocatalytic applications for VOCs degradation. Specifically, the synthesis methodologies of BiVO_4_-based materials, the engineering of BiVO_4_ to achieve changes in its structural and electronic properties, and the correlation of these properties with improvement in visible light photocatalytic activity are also discussed. Finally, further prospects for BiVO_4_-based materials are presented, which may provide a better understanding and encourage the field-scale application of BiVO_4_-based materials for VOCs degradation.

## 2. Advanced Oxidation Process and Semiconductor Photocatalysis

Advanced oxidation processes (AOPs) are amongst the widely accepted eco-friendly processes for the treatment of different wastewaters. They involve the in-situ generation of hydroxyl (^•^OH) and sulphate (SO_4_^•^) radicals, which are strong oxidants for the oxidation of various types of toxic organic pollutants. Among these, the hydroxyl (^•^OH) radical is the most efficient species in AOPs. Some AOPs, including photolysis (UV) and photochemical (UV/H_2_O_2_, UV/O_3_) reactions, the Fenton reaction (Fe^2+^/H_2_O_2_), photo-Fenton reactions (light/Fe^2+^/H_2_O_2_), cavitation (ultrasonic irradiation), and electrochemical and photocatalysis have been utilized effectively for the oxidation of pollutants [44,45]. Among these, photocatalytic processes have been effectively utilized in a series of oxidation and reduction reactions on the surface of semiconductor materials in the presence of light irradiation. A Web of Science bibliometrics (Figure 1) analysis showed that photocatalytic oxidation was an efficient process among AOPs for the degradation of hazardous pollutants, especially volatile organic compounds (VOCs).

When the semiconductor absorbs the photon at not less than the band gap energy of the semiconductor (E_g_), the electrons (e^−^) from the valence band (VB) are excited towards the conduction band (CB), and holes (h^+^) are left behind in the VB. The photogenerated electrons and holes pairs move into the surface of the semiconductor where they react with surface-adsorbed water or hydroxyl (^−^OH) groups or dissolved oxygen in the reaction medium and produce reactive radical species, i.e., superoxide radical anions (O_2_^•−^) and ^•^OH radicals. The reactive radical species undergo redox reactions with surface-adsorbed pollutant molecules and completely degrade them. Furthermore, the presence of holes in the VB directly oxidizes the surface-adsorbed pollutants and electrons in the CB, contributing to the indirect oxidation of pollutants by ^•^OH radicals generated through the photo-splitting of hydrogen peroxide (H_2_O_2_) formed in situ. The corresponding steps involved in the photocatalytic oxidation process using semiconductor materials, and their schematic representation, are shown in Equations (1)–(12) and Figure 2.
Semiconductor + *hυ* (λ ≥ band gap energy) → Semiconductor (h^+^) + Semiconductor (e^−^)(1)
h^+^ + H_2_O → Semiconductor + H^+^ + ^•^OH(2)
Semiconductor (h^+^) + -OH → Semiconductor + ^•^OH(3)
Semiconductor (e^−^) + O_2_ → Semiconductor + O_2_^• −^(4)
O_2_^• −^+ H^+^ → HO_2_^•^(5)
HO_2_^•^ + e^−^ → HO_2_^−^(6)
HO_2_^−^ + H^+^ → H_2_O_2_(7)
HO_2_^•^ + HO_2_^•^ → H_2_O_2_ + O_2_(8)
H_2_O_2_ + *hυ* → 2 ^•^OH(9)
H_2_O_2_ + Semiconductor (e^−^) → Semiconductor + OH^−^ + ^•^OH(10)
Semiconductor (h^+^) + pollutants → CO_2_ + H_2_O + other products(11)
^•^OH + O_2_^•−^ + VOCs → intermediate products → CO_2_ + H_2_O + other products(12)

Initially, various metal oxides (TiO_2_, ZnO, WO_3,_ etc.) and metal sulfide-based (e.g., ZnS, CdS, etc.) semiconductor materials were utilized as catalysts for the photocatalytic oxidation of pollutants, as shown in Table 1. TiO_2_ is the best among the semiconductor materials as it has the characteristics of chemical and biological stability, low toxicity, high durability, resistance to photocorrosion, high photocatalytic activity, and low cost. Although TiO_2_ has these merits, it has some disadvantages, such as high bandgap energy (~3.2 eV), poor adsorption capacity, low surface area, and a high recombination rate of photogenerated charge carriers that limit the practical applicability of TiO_2_ materials. Furthermore, the high bandgap energy restricts its usage under simulated or natural solar light irradiation, and the high recombination rate of charge carriers reduces the photocatalytic degradation efficiency. Approaches such as metal or non-metal doping, coupling with high surface area adsorbents (e.g., activated carbon, graphene oxide, etc.) and other semiconductors (e.g., WO_3_, g-C_3_N_4_, etc.), reduction in size of the semiconductor, changes to the morphology and dye sensitization, etc., have been used to enhance the visible light response and the lifetime of photogenerated charge carriers, consequently improving the photocatalytic degradation efficiency of TiO_2_ materials. Nevertheless, modified TiO_2_ materials are limited in their industrial application because of the poor stability (leaching or natural degradation) and reusability of the materials, the high cost of dopants, harmful modifying agents, and poor re-production/re-synthesis of material with similar properties. In order to overcome this, various non-TiO_2_-based visible-light-responsive semiconductor materials have been developed for the photocatalytic oxidation of various pollutants, including VOCs.

Among non-TiO_2_ semiconducting materials, bismuth-based semiconducting materials [BiVO_4_, Bi_2_WO_6_, Bi_2_MoO_6_, BiOX (X–Cl, Br, I), etc.,] have received much attention because of their narrow band gap energy, excellent visible light absorption, considerable chemical and thermal stability, and suitable band potentials for the generation of reactive radical species [46,47]. BiVO_4_ is one of the best bismuth-based semiconducting materials due to its low band gap energy (E_g_ = 2.3–2.4 eV) and high visible light absorption. The discovery of BiVO_4_ by Kudo et al. [43] for O_2_ evolution under visible light irradiation has significantly influenced the development of BiVO_4_-based materials for various applications, including VOCs degradation. Nevertheless, the faster rate of photogenerated electron–hole pair recombination, poor charge carrier transport ability, and the water oxidation kinetics of BiVO_4_ limit its industrial application. A large number of modifications have been performed to improve the photocatalytic performance of BiVO_4_ so that commercial requirements would be fulfilled. These are described in the forthcoming sections.

## 3. Fundamental Aspects of BiVO_4_ Photocatalyst

BiVO_4_ is an n-type semiconductor and has been identified as one of the most efficient visible-light-responsive photocatalysts with a band gap energy of 2.4 eV. Naturally, BiVO_4_ occurs as the pucherite mineral with an orthorhombic crystal structure. However, laboratory-prepared BiVO_4_ crystallizes either in a scheelite or zircon-type structure. Furthermore, the scheelite structure has a monoclinic and tetragonal crystal system, and the zircon-type structure has a tetragonal crystal system [48]. The crystal structures are shown in Figure 3.

Similarly, the band structures of scheelite and zircon-type BiVO_4_ are shown in Figure 4. In zircon-type BiVO_4_, the valence and conduction bands are comprised of O 2p and V 3d orbitals, whereas in the case of scheelite-type BiVO_4_, the valence band consists of Bi 6s and O 2p orbitals and the conduction band consists of a V 3d orbital. Therefore, the monoclinic (m-BiVO_4_) scheelite-type system has a relatively smaller band gap energy (2.4 eV) than the tetragonal zircon-type system (2.9 eV); therefore, it shows high visible-light-driven activity [50].

However, the utilization of BiVO_4_ for catalytic activity is not impressive because it suffers from a high recombination rate of electron–hole pairs and poor charge transport properties. Therefore, the modification of BiVO_4_ materials, such as by doping of BiVO_4_ with metals and non-metals, to control its morphology and the synthesis of composite BiVO_4_ (heterojunction, S-scheme, and Z-scheme) materials has been performed to overcome the abovementioned shortcomings, which are discussed in forthcoming sections.

## 4. Synthesis Methodologies of BiVO_4_ Photocatalyst

It is clear that the crystallinity, particle size, and shape of photocatalysts have a significant impact on the photocatalytic activity of a catalyst. Furthermore, it is well known that photocatalytic processes take place on the photocatalyst’s surface. A substantial increase in a photocatalyst’s surface-to-volume ratio will increase its specific surface area, increasing the number of active sites that are available for photocatalytic reactions [50,51]. The morphology and particle size are directly proportional to the greater surface recombination of photogenerated charge carriers. The photocatalyst produces electrons and holes in the diffusion time; the diffusion time of the photocatalyst from the bulk to the surface is represented by Equation (13).
τ = r^2/^π^2^ D(13)
where r is the grain radius and D is the diffusion coefficient of the charge carrier. Therefore, when the grain radius decreases, a large number of electrons and holes will travel to the surface for photocatalytic reaction. The synthesis of small and uniform particle sizes plays a vital role in enhancing the photocatalytic activity of semiconducting materials [49,50,51]. The various synthesis methodologies of BiVO_4_ with different morphologies and their photocatalytic activity and degradation efficiency are summarized in Table 2.

### 4.1. Hydrothermal Method

Kamble and Ling synthesized truncated square, 18-sided morphological BiVO_4_ nanomaterials using the hydrothermal method. Figure 5a–d show the different sizes and shapes of m-BiVO_4_ nanoparticles (NPs). Figure 5c shows the truncated square (18-sided) hexagonal bipyramidal shape with exposed {040} facets exhibiting a strongly revealed surface phenomenon and a facet effect for the visible-light-driven photocatalytic degradation of MB dye [42].

Sun et al. [51] synthesized a BiVO_4_ nanoplate-stacked star morphological structure via a hydrothermal method using ethylenediamine tetraacetic acid (EDTA). The molar ratio of EDTA to Bi^3+^ played an important role in the star morphology of BiVO_4_. The star-like BiVO_4_ structure showed a higher efficiency for the photodegradation of MB in 25 min under visible light irradiation. The dendritic structure of BiVO_4_ was fabricated by Lei et al. [52] using an additive-free hydrothermal method at various hydrothermal temperatures, including 100 °C, 140 °C, and 180 °C (Figure 6). The synthesized dendritic structure of BiVO_4_ was used as a photocatalyst for the degradation of RhB dye. The dendritic BiVO_4_ synthesized at 140 °C (99.3%) showed superior photocatalytic activity to material synthesized at 100 (58.3%) and 180 °C. The higher activity may be attributed to the high surface area (2.1 m2/g) and crystallinity compared to the other BiVO_4_ samples.

An olive-like BiVO_4_ hierarchical morphology via a template-free hydrothermal method was synthesized by Wang et al. [53]. The photocatalytic activities of olive-like BiVO_4_ was estimated against MB under visible light irradiation. The BiVO_4_ hierarchical morphology was prepared at different pH values. At a pH of 2.05, the BiVO_4_ appeared as an olive-like structure but steadily changed into a spherical structure when the pH value varied from 2.05 to 4.02, and at pH 6.00, it adopted a cuboid structure. The authors reported that the olive-like BiVO_4_ resulted in ∼95.7% degradation of MB within 1 h under visible light conditions.

S. Obregon et al. prepared BiVO_4_ hierarchical heterostructures using a surfactant-free hydrothermal method [54]. The m-BiVO_4_ prepared at pH 9 showed needle-like morphology with {110} and {002} planes. The synthesized m-BiVO_4_ showed good photoactivities for the degradation of methylene blue (MB) under UV–Vis irradiation.

Lu et al. [55] fabricated core–shell-structured (CSS) BiVO_4_ via a surfactant- and template-free hydrothermal method using bismuth nitrate/ammonium vanadate/ethanol/acetic acid as precursors. They also synthesized different morphologies using PVP and CTAB surfactants. Figure 7a–f show that the shell of the BiVO_4_ hollow spheres became thinner as the hydrothermal reaction time increased. The BiVO_4_ plate morphology and biscuit morphology showed ∼88% degradation of RhB at a very slow rate, while the BiVO_4_ biscuit morphology with a core shell structure showed ∼99% degradation of RhB in 4.5 h.

Meng et al. [56] fabricated various nanoparticles with polyhedral, rod-like, tubular, leaf-like, and spherical morphological structures using a hydrothermal method in the presence of triblock copolymer P123 as a surfactant (Figure 8). Their photocatalytic activities were estimated towards the decomposition of MB dye under visible light stimulation. At different pH = 1, 6, 9, or 10 conditions, the 2D nanoentities were annealed at 400 °C and showed the abovementioned different morphologies. Among the various BiVO_4_ morphologies, those synthesized hydrothermally with P123 at pH 6 or 10 showed excellent photocatalytic activity due to their greater surface areas and high concentrations of surface oxygen defects. Two-dimensional BiVO_4_ single-crystal nanosheets were prepared by Zhang et al. [57] with thicknesses of ∼10–40 nm via a hydrothermal route using sodium dodecyl benzene sulfonate (SDBS) as an anionic surfactant. SDBS formed micelles in aqueous solution and enabled the synthesis of BiVO_4_ nanoparticles with controlled growth. The as-synthesized BiVO_4_ nanoparticles were used for the photocatalytic degradation of the RhB dye.

A monoclinic BiVO_4_/sepiolite nanocomposite was fabricated by H. Naing et al. [58]. The nanocomposite BiVO_4_/sepiolite exhibited excellent visible light photocatalytic performance against antibiotic tetracyclines (TCs) and methylene blue (MB). In the monoclinic BiVO_4_-30%-sepiolite, fibrous or needle-like sepiolite was distributed on the peanut-shaped monoclinic BiVO_4_ surface. The photocatalytic efficacies of pure BiVO_4_, pure sepiolite, and serial monoclinic BiVO_4_/sepiolite (BVO/S) nanocomposites were studied using the remediation of MB dye and TCs in aqueous solution under visible light irradiation. About 96% of the MB pollutants and 78% of the antibiotic TCs were degraded by BVO-30% S after 4 h of visible light irradiation. A synergistic effect between sepiolite and monoclinic BiVO_4_ enhanced the separation of the photo-generation carriers, promoting high adsorption, and restrained the regrouping of electron–hole pairs, enhancing photocatalytic activity. Moreover, the hydrophobic nature of sepiolite nanofiber possibly enabled holes generated on the BiVO_4_/sepiolite nanocomposites to react with pollutants and degrade to smaller molecules.

Chen et al. [59] synthesized snow-like BiVO_4_ using a cetyltrimethylammonium bromide (CTAB)-assisted hydrothermal method. In the snow-like BiVO_4_ morphology, oxygen vacancies depended upon the concentration of CTAB. The snow-like BiVO_4_ morphology coexisted with counter-Br^−^ ions, inducing high-concentration surface oxygen defects, which produced more highly reactive oxygen species (ROS), i.e., superoxide and hydroxyl radicals, which resulted in the superfast degradation of ciprofloxacin (CIP).

### 4.2. Electro-Spinning Method

Various 1D nanostructured BiVO_4_ materials, such as nano-fibers [60] and micro-ribbon [61], have been synthesized for photocatalysis applications. Cheng et al. [60] prepared BiVO_4_ porous 1D nanofibers by an electro-spinning method using polyvinyl pyrrolidone (PVP)/acetic acid/ethanol/N, N-dimethylformamide/bismuth nitrate/vanadium (IV) oxy acetylacetonate as a precursor, and the photocatalytic efficiency was estimated towards the photodegradation of RhB dye. The authors reported that the 500 °C calcined BiVO_4_ sample displayed a higher photocatalytic efficiency for RhB than that for other calcinating temperatures. Liu et al. [61] reported micro-ribbon BiVO_4_ of an ∼2–3 μm width for the visible-light-driven photocatalytic degradation of MB dye. They also studied the impact of a calcinating temperature of 500 °C on the morphology of BiVO_4_.

### 4.3. Solvothermal Method

Red blood cell, flower-like microsphere and dendrite BiVO_4_ morphologies were prepared by Chen et al. [62] via a facile solvothermal method by adjusting the solution pH and using bismuth nitrate/ammonium vanadate/citric acid/ethylene glycol/ethanol/water as precursors. Figure 9 illustrates the various morphologies and microstructure of the BiVO_4_ samples using FE-SEM micrographs. Figure 9a,b show the morphology of BiVO_4_ red blood cell (S-BiVO_4_), which was achieved using Na_2_CO_3_ as a pH-controlling agent. The flower-like microsphere BiVO_4_ nanoparticles (A-BiVO_4_) were obtained using NH_3_·H_2_O, as shown in Figure 9c,d. Figure 9e shows the BiVO_4_ dendrite-like morphology (N-BiVO_4_), which was achieved without the addition of citric acid and Na_2_CO_3_ under similar conditions. The BiVO_4_ red blood cell (S-BiVO_4_) catalyst exhibited greater catalytic activity than the flower or dendritic morphologies of BiVO_4_.

### 4.4. Co-Precipitation Method

Mesoporous monoclinic BiVO_4_ photocatalysts with different morphologies were prepared by Suwanchawalit et al. [63]. In the synthesis of m-BiVO_4_, TX100 was used as a surfactant in the co-precipitation method. The TX100 molecules play an important role in the synthesis of BiVO_4_. An m-BiVO_4_ structure was prepared by Lai et al. [64] via a precipitation method using a visible light catalyst for the photocatalytic degradation of thiobencarb (TBC). TBC was efficiently degraded by approximately 97% within 5 h. The as-prepared BiVO_4_ photocatalyst had a polyhedral morphology with a 6–8 μm edge length (Figure 10).

### 4.5. Sol–Gel Method

The sol–gel method is a promising approach for synthesising metal oxide/mixed oxide composites as this methodology is capable of controlling the morphological and surface properties of the materials at the nanoscale [65]. Min et al. synthesized La- and B-doped BiVO_4_ photocatalysts using the sol–gel method and utilized them for the photocatalytic degradation of MO dye [66]. Co-doped BiVO_4_ photocatalysts may enable the synergistic effects of lanthanum and boron to separate the photogenerated holes and electrons in BiVO_4_ composite. Although all synthesized bismuth vanadate composites have spherical structures, some La-doped composites show a decrease in particle size, and La-doping also inhibits particle growth. In this study, due to La and B doping, BiVO_4_ (La-B-BiVO_4_) composites showed a higher photocatalytic degradation of MO dye in 60 min than BiVO_4_ and B-BiVO_4_, and the specific surface area and surfaces for oxygen vacancies were also enhanced, reducing the crystallite size, and also reducing the band gap and the intensity of absorbed light in the visible region.

Mousavi-Kamazani synthesized composite nanostructures of copper oxides and bismuth (Cu/Cu_2_O/BiVO_4_/Bi_7_VO_13_) using the Pechini sol–gel method [67]. By altering the reaction conditions, different morphological structures were synthesised. The addition of ethylenediamine as a gelling agent, tannic acid as a chelating agent, and a 1:1:1 ratio for Cu:Bi:V enabled a rectangular cube-like morphology to be formed. When the gelling agent changed to polyethene glycol instead of ethylenediamine, plate-like microstructures were formed. By changing the chelating agent from tannic acid to fumaric acid, a pseudo-spherical morphology was obtained. In the absence of Cu and 0.5 mole Cu, platelike nanostructures, nanorods, and quasi-spherical structures were observed, respectively. Moreover, the composite structure (Cu/Cu_2_O/BiVO_4_/Bi_7_VO_13_)) with rectangular cube-like morphology (size of about 30–100 nm) exhibited excellent photocatalytic oxidative desulfurization of the oil derivatives under visible light (92%) than BiVO_4_ and Cu_2_V_4_O_11_. The results suggest that the addition of Cu and Cu_2_O species into the composites increases the electrical conductivity, capable of electron–hole separation, and alters the morphology and also particle size, which might be the reason for the enhanced photocatalytic activity.

Castaneda et al. synthesized BiVO_4_/TiO_2_ nanocomposites with different compositions (BiVO_4_: TiO_2_ = 1:0.6, 1:2.5, and 1:10) using a modified one-step sol–gel method [68]. The SEM studied before and after calcination shows diversity in the particle size and exhibits a spherical shape. The BiVO_4_/TiO_2_ nanocomposite with a mass ratio of (1:10) shows the highest photocatalytic efficiency compared to the other compositions and (photo) electrochemical responses for the degradation of azo dyes (Acid Blue-113, AB-113) (~99%) under visible light radiation.

## 5. Engineering/Modification Processes of BiVO_4_ Properties

As discussed earlier, BiVO_4_ is one of the promising photocatalysts for many practical applications ranging from water treatment, the removal of dyes and organic pollutants, H_2_ generation, cargo and biomedical deliveries, etc. [69,70]. The better photocatalytic performance of BiVO_4_ is accredited to their visible-light-responsive band gap (2.4 eV), layered structure, suitable valance band maximum, chemical stability, and nontoxicity. BiVO_4_ has four different polymorphic forms, including *orthorhombic, zircon-tetragonal, monoclinic (m),* and *tetragonal (t)* BiVO_4_ [71]. Among them, m-BiVO_4_ is more active for photo-related applications. Generally, a low temperature reaction yields zircon-tetragonal BiVO_4_ phase, which can be transformed into m-BiVO_4_ by inducing calcination reactions and reversibly back to t-BiVO_4_ phase at a temperature of 528° K. The band gaps energies of t-BiVO_4_ and m-BiVO_4_ were reported to be indirect of 2.3 eV and 2.4 eV, respectively [72]. Previous experimental studies with BiVO_4_ have some shortcomings, such as poor charge carrier transfer (bulk carrier mobility of 0.05–0.2 cm^2^ V^−1^s^−1^) and charge recombination before being captured by targeted molecules for photochemical reactions [73]. To overcome these issues, several strategies, for example, the nano-scaling [74], morphology engineering [75], crystal facet control [76], and crystal structure control [77] of BiVO_4_, have been demonstrated to improve the optical and electronic properties to some extent, resulting in high photocatalytic performances towards organic pollutant degradation.

Many recent studies have also proposed multiple advantages of doping and mixed-phase BiVO_4_ systems over single-phase BiVO_4_ photocatalysts [78,79,80]. This resulted in the extended light absorption capability, carrier mobility, and higher efficiency of BiVO_4_-based systems for photochemical reactions. This is mainly attributed to the effect of donor defects or through adding excess electrons to the BiVO_4_ model system through doping with metals (Mo, W, Sn) [69,81,82], nonmetals (S, F), and the creation of heterostructures with other semiconductors. In this section, an overview of different strategies such as (1) metal doping, (2) noble metal doping, (3) nanocomposite structures, (4) composite with carbon analogs, and (5) heterostructures related to BiVO_4_ photocatalyst have been briefly discussed.

### 5.1. Metal/Nonmetal-Doped BiVO_4_

Metal doping is one of the conventional and effective strategies for modifying the electronic properties of semiconductors, i.e., p- and n-type conductivity behaviors. In semiconductor photocatalysis, doping with metal/nonmetal can improve the charge carrier separation, tune the band gap energy, and enhance visible light absorption. For example, Liu et al. [83] synthesized Mo-doped BiVO_4_ using an electrospun method and studied its morphology, crystal structure, and optoelectronic properties. A stoichiometric amount of bismuth and vanadium solutions were prepared, and varied amounts of ammonium molybdate (0.4, 1, 1.5, 3 mol%) were incorporated and stirred (12 h) to obtain homogeneous solution. Then, an electrospinning reaction was performed at a temperature of 60 °C and 15 kV voltages. According to the formation mechanism (Figure 11A(i)), pure BiVO_4_ can form homogenous, well-dispersed particles together with many pores. When a small amount of Mo (1%) was incorporated into BiVO_4_, the particle size increased, and the existing pores disappeared (Figure 11A(iii–v)). When further increasing the Mo content (3%), the BiVO_4_ particle size increased and Mo saturated, forming secondary-phase MoO_3_ on the surface of BiVO_4_. As a result, photocatalytic tests revealed that 1% Mo-BiVO_4_ shows excellent photoactivity ~ three times higher than reference BiVO_4_ (Figure 11A(ii)), indicating that a small content of Mo dopant is crucial to improving the separation of charge carriers and electronic conductivity.

On the contrary, a higher Mo content led to crystal phase transformation from monoclinic to tetragonal together with secondary-phase formation, which might act as recombination centers causing a reduction in the photocatalytic activity. A theoretical study by Zhang et al. [84] purports the effects of Mo/W co-doping for the photocatalytic activity of monoclinic BiVO_4_. They found that Mo or W atom doping preferably occurs at the V site to generate continuum states directly above the conduction band (CB) level of BiVO_4_, and this decreases the band gap, which is beneficial for photochemical reactions. Particularly, they found that W-doped BiVO_4_ exhibits a smaller band gap than the Mo-doped BiVO_4_, and the electronic properties of BiVO_4_ are quite different. Additionally, Mo/W/Mo and W/Mo/W co-doping in BiVO_4_ requires low formation energies and reduced bandgaps compared to other doping systems, which may extend the light absorption and could be more suitable for visible-light-driven photocatalysis. Yao et al. [85] fabricated Mo-doped BiVO_4_ via a solid-state reaction by grinding stoichiometric amounts of Bi_2_O_3_, V_2_O_5_, and MoO_3_ in an agate mortar and heating at 600◦C for 5 h, followed by calcination at 800 °C for 2 h. The Mo-doped BiVO_4_ showed much more photocatalytic activity for water oxidation and MB dye degradation compared to pure BiVO_4_, due to the higher surface acidity of Mo-doped BiVO_4_ (2 atom%) stemmed from the existence of Lewis and Brønsted acidic sites associated with Mo^6+^ doping, which offer greater adsorption feasibility for water molecules and organics contaminants. Gao et al. [86] reported the synthesis of Ni-doped BiVO_4_ and Z-scheme BiVO_4_-Ni/AgVO_3_ nanofibers using a strategy which combined an electrospinning and hydrothermal strategy (Figure 11B(i)). First, a nanofibrous Ni-doped BiVO_4_ was obtained using an electrospinning precursor solution of Bi(NO_3_)_3_⋅5H_2_O, Vo(acac)_2_, and Ni(NO_3_)_2_⋅6H_2_O (prepared in mixed solvent of DMF, CH_3_COOH, and CH_3_CH_2_OH). PVP was used a matrix during the electrospinning process. Next, hydrothermally grown AgVO_3_ on the surface of Ni-doped BiVO_4_ forms a Z-scheme heterojunction of BiVO_4_-Ni/AgVO_3_. Due to the synergetic effect of Ni doping and AgVO_3_ assembly, the specific surface area and light absorption ability of BiVO_4_-Ni/AgVO_3_ was significantly improved compared to BiVO_4_. Ni-doping adds impurity energy levels and replaces V sites on the {121} plane (Figure 11B(ii)), producing the structural distortion of tetrahedral VO_4_^3+^, which can be confirmed from a right shift in diffraction peaks [87]. The lowering of the diffraction peak intensity also suggests successful Ni doping, which leads to a reduction in crystallinity. A Z-scheme optimal BiVO_4_-Ni-1/AgVO_3_-25 photocatalyst showed superior photocatalytic Cr^6+^ reduction efficiency (99.7%) in 80 min. The comparison of apparent rate constants for Cr^6+^ reduction over different photocatalysts is shown in Figure 11B(iii). The inset shows the HRTEM image of BiVO_4_-Ni/AgVO_3_ and the presence of three different phases.

**Figure 11 nanomaterials-13-01528-f011:**
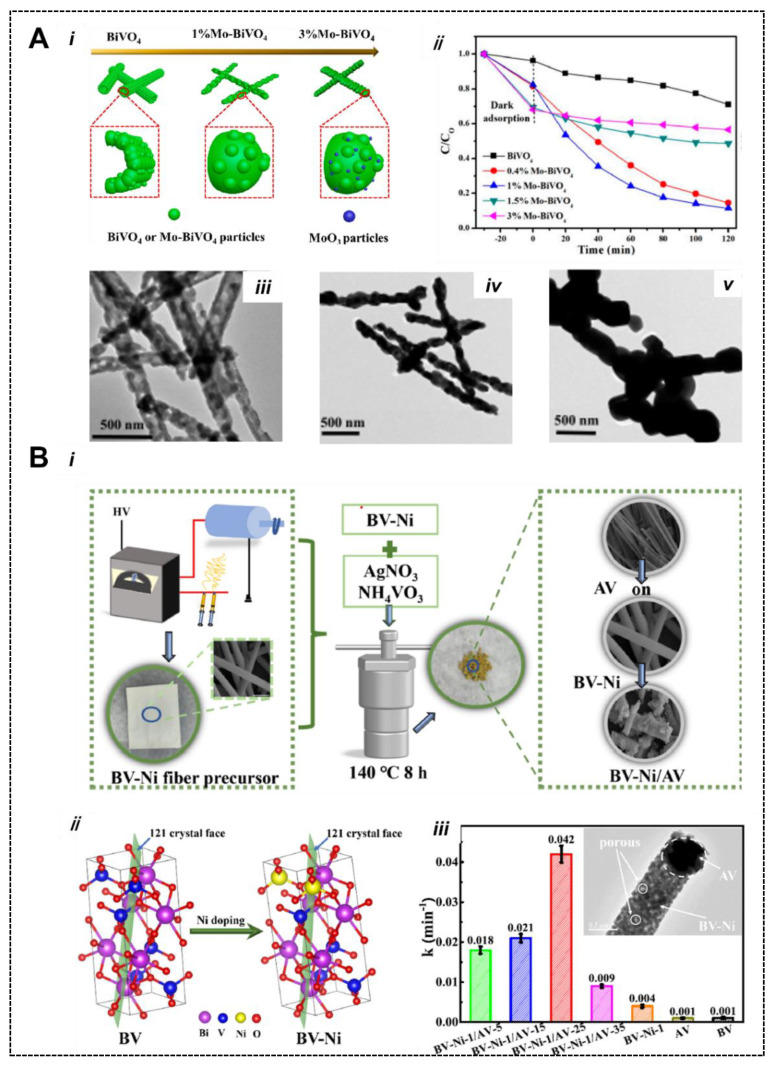
(**A**) (**i**) Schematic showing the electrospinning synthesis strategy for Mo-doped BiVO_4_ nanostructures. (**ii**) Photocatalytic degradation of MB using Mo−BiVO_4_ photocatalyst under light irradiation. (**iii**–**v**) TEM image of pure BiVO_4_, 1% Mo-BiVO_4_, and 3% Mo-BiVO_4_ (reproduced with permission from Ref. [83]). (**B**) (**i**) Synthetic protocol for nanofibrous BiVO_4_-Ni/AgVO_3_ [86], (**ii**) schematic representation of possible doping sites of Ni in BiVO_4_, (**iii**) photocatalytic reduction of Cr^6+^ over different photocatalysts and their apparent rate constants and inset shows the TEM image of BV-Ni-1/AV−25. Reproduced with permission from Ref. [87].

Bashir et al. [88] fabricated Gd-doped BiVO_4_ using a simple hydrothermal method. The resulting Gd-doped BiVO_4_ ultrasonically treated with rGO to form Gd-doped BiVO_4_/rGO of the nanocomposite nanostructures (Figure 12A(i)). According to SEM analysis, Gd doping does not change the surface morphology. As shown in TEM images (Figure 12A(ii,iii)), three different phases of components promote significantly improved electron/hole pair separation, excellent photocatalytic MB degradation efficiency (97%) than BiVO_4_ (53%), and Gd/BiVO_4_ (69%) within 100 min (Figure 12A(iv)). The higher photoactivity of rGO/Gd/BiVO_4_ is due to a heterojunction effect between Gd/BiVO_4_ and rGO sheets, which not only enhances the light absorption but also enlarges the surface area in the presence of rGO.

Unlike transition metals, doping with a rare earth metal is also found to improve the photocatalytic properties; however, this finding is rarely reported [89]. These metal ions possess excellent luminescence properties and therefore endow several benefits such as light absorption, modified surfaces, and acting as electron traps that can help minimize the recombination of photoinduced charge carriers [90]. Moscow et al. [91] reported erbium (Er) and yttrium (Y)-doped BiVO_4_ using a simple microwave-assisted approach (Figure 12B(i)). In synthesis, Bi and V precursors were dissolved under magnetic stirring. Later, different amounts of Er and Y precursors were introduced, followed by microwave irradiation forming Er^3+^- and Y^3+^-doped BiVO_4_ nanostructures. According to SEM and XRD results, Er^3+^ and Y^3+^ doping led to a reduction in the particle sizes of BiVO_4_ and formed mixed-phase BiVO_4_. Raman spectra analysis revealed (V–O) band shift from 820 cm^−1^ to 850 cm^−1^ and disappeared δ (VO_4_^+^) doublet due to the conversion of monoclinic BiVO_4_ to tetragonal BiVO_4_ phase (Figure 12B(ii)). Photocatalytic tests suggested that Y-doped BiVO_4_ reportedly has the highest degradation efficiencies of 93%, 85%, and 91% for MB, MO, and RhB dyes, respectively, in 180 min under light irradiation. The possible photocatalytic electron transfer mechanism is shown in Figure 12B(iii). A photocatalytic degradation of acetaldehyde was also achieved at an impressive rate using Y-doped BiVO_4_. Because of the formation of the inner energy state Er^3+^ and Y^3+^ metal, the band gap reduced, light absorption extended, and the recombination of electron–hole pairs suppressed. Next, the co-doping of Gd and Y into BiVO_4_ was successfully accomplished by simple hydrothermal synthesis [92]. Upon sunlight illumination (90 min), a Bi_0.92_Gd_0.07_Y_0.01_VO_4_ photocatalyst exhibited 94% degradation efficiency for methylene blue dye (MB), which is four times larger than pure BiVO_4_. Recently, Sudrajat and Hartuti [93] used a one-step hydrothermal preparation method to prepare B-doped BiVO_4_ (B-BiVO_4_) with an oval-shaped morphological structure. The B dopant acts as mid-gap-state electron donors, allowing more excitations of the band gap to be produced and the conduction band of BiVO_4_. The light-induced infrared absorption measurement confirmed that there were more electrons available for reduction reactions and more holes available for oxidation reactions, leading to greater photocatalytic activity for the mineralization of phenoxyacetic acid (PAA) in the presence of simulated sunlight.

**Figure 12 nanomaterials-13-01528-f012:**
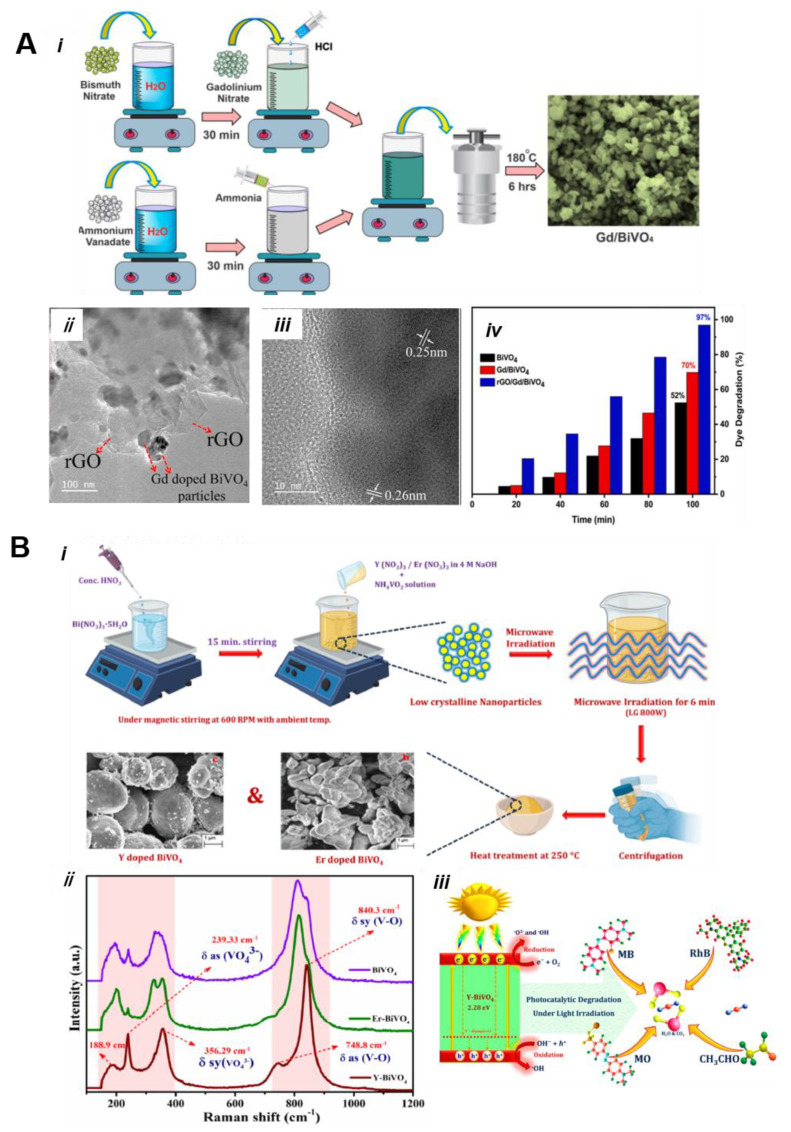
(**A**) Gd-doped BiVO_4_. (**i**) Schematic illustration of the synthesis strategy, (**ii**) TEM images of rGO/Gd/BiVO_4_, (**iii**) HRTEM image of Gd/BiVO_4_, and (**iv**) comparison of dye removal efficiency with different photocatalysts (reprinted with permission from Ref. [88]). (**B**) Y- and Er-doped BiVO_4_. (**i**) Schematic representation of microwave-assisted synthesis, (**ii**) Raman spectra of BiVO_4_, Y-BiVO_4_, and Er-BiVO_4_, and (**iii**) schematic showing various dye degradation using Y−BiVO_4_ photocatalyst with sunlight. Reprinted with permission from Ref. [91].

Similarly, doping with nonmetals such as nitrogen (N) and fluorine (F) in BiVO_4_ has been adopted to promote light absorption, band gap tuning, and catalytically active surfaces. For instance, Wang et al. [94] prepared N-doped BiVO_4_ using a sol–gel technique with hexamethylene tetramine (C_6_H_12_N_4_) as a N source. N-doping was found to not change the morphology and surface area of BiVO_4_ significantly. However, N was found to be doped into crystal lattice O–Bi–N–V–O bonds, creating highly active V^4+^ species, oxygen vacancies, and a red shift in the absorption band. The photoactivity of BiVO_4_ in this system mainly depends on two factors: (i) the content N-doping and (ii) heat treatment temperature. The N-BiVO_4_ calcined at a temperature of 500 °C showed the highest activity. Li et al. [95] reported the synthesis of F-doped BiVO_4_ nanospheres via a simple two-step hydrothermal method. NaF was employed as the fluoride source which helped to modify the crystal lattice of BiVO_4_, suppressing the charge carrier recombination, resulting in high photoactivity. Wang et al. [96] synthesized B-doped-BiVO_4_ photocatalysts using a CS-template-assisted sol–gel method. According to these authors, B doping can form a monoclinic crystal structure, large surface areas, a smaller band gap value, and higher V^4+^ species. This results in the best photoactivity of 0.04B CS-BiVO_4_ for the degradation of MO dye, which is not only because of B doping but also due to the cellular morphology, which stems from the template as a similar 0.04B-BiVO_4_ sample prepared without a template showed much lower photoactivity.

### 5.2. Noble-Metal-Doped BiVO_4_ as Photocatalyst

Noble metals (Au, Ag, Pt) are very attractive in photocatalysis because of their surface plasmon resonance (LSPR), which can provoke the oscillation of conduction band electrons and plasmonic energy transfer. This mainly occurs via two different mechanisms: (i) direct electron transfer and (ii) plasmon-directed resonant transfer of energy. Cao et al. [97] have reported the synthesis of Au-BiVO_4_ nanosheets using hydrothermal and a cysteine-linking strategy. Au precursor in the presence of cysteine evolves Au-doped BiVO_4_, as shown in Figure 13A(i). Interestingly, the surface plasmon resonance (SPR) of Au enables excellent visible-light-driven photocatalytic activity related to pure BiVO_4_ for the degradation of MO dye (Figure 13A(ii)). In this system, Au acts as an electron sink retarding the recombination of photoinduced electrons and holes. The contribution from Au nanoparticles was clarified by comparing the experimental results with Pt-BiVO_4_ (prepared through a similar strategy), showing no SPR in the range of (500 ± 20 nm) as Au-BiVO_4_ under visible light illumination. Moreover, electron trapping on Au raises the Fermi level (E_f_) of Au to more negative potentials (E_f_*), leading to band alignment for an effective charge transfer. In addition, the photogenerated electrons transfer to Au nanoparticles can reduce the adsorbed sacrificial agent S_2_O_8_^2−^ to SO_4_^2−^ on the Au surface. As a result, the holes remain on the BiVO_4_ surface and have a considerably higher lifetime to perform the water oxidation process (Figure 13A(iii)). Reddy et al. [98] reported Au-doped BiVO_4_ photocatalyst synthesis via a sonication and calcination method and applied it as electrodes for water splitting and electrochemical storage. This study suggests that, due to Au doping, the photocurrent density increased 25 times related to reference BiVO_4_, demonstrating the synergistic role of Au, while BiVO_4_ increased the electrical conductivity and charge transfer at the interface. The synergistic effects of Ag nanoparticles (LSPR) and N-doped graphene (upconversion effect) with BiVO_4_ have also been reported for the degradation of tetracycline hydrochloride (TC•HCl), as shown in Figure 13B(i) [99]. A ternary photocatalyst N-GNDs/Ag/BiVO_4_ was prepared through the solvothermal and hydrothermal process, and it exhibits distinct behavior from reference BiVO_4_. According to the experimental results, possible charge transfer mechanisms using N-GNDs/Ag/BiVO_4_ have been schematically proposed (Figure 13B(ii)). Under light illumination, N-GNDs can absorb NIR light and contribute to the light upconversion phenomenon, which can help to extend light absorption. Since the potential of O_2_/O_2_^−^ (−0.33 eV, NHE) is a higher negative value than the conduction band of BiVO_4_ (+0.47 eV, NHE), the conduction band electron of BiVO_4_ cannot reduce O_2_ to produce ^•^O_2_^−^ radicals. As a result, the photoinduced electron from BiVO_4_ migrates to N-GNDs and Ag-NPs, leading to band alignment. Thus, a Schottky barrier is formed at the interface to facilitate more efficient electron transfer; N-GNDs serve as electron acceptors to capture photoexcited electrons from BiVO_4_, and hot electrons from Ag-NPs (LSPR effect) reduce O_2_ to ^•^O_2_^−^, which later oxidizes TC•HCl to smaller molecules.

### 5.3. BiVO_4_-Based Nanocomposites

Single-phase BiVO_4_ is insufficient to achieve higher photocatalytic efficiency due to the poor electron transfer and low light absorption ability (band gap 2.4 eV). Recently, a concept of nanocomposite fabrication has been adopted, in which BiVO_4_ is coupled with other metals such as Au [98], Ag [100], Co [101], semiconductors TiO_2_ [102], ZnO [103], CdS [104], WO_3_ [105], MnO_2_ [106], Ag_3_VO_4_ [107], and carbon analogues graphene [99,108] g-C_3_N_4_ [109], carbon nanotubes [110], etc., enabling band gap alignment and efficient electron transfer across interfaces. The resulting composites could improve the overall efficiency by modifying the chemical, physical, and optical properties of BiVO_4_ and charge carriers and light absorption.

Wei et al. [111] fabricated N-doped Biochar (N-Biochar)@BiVO_4_ nanocomposite via an easy hydrothermal method and evaluated its photocatalytic performance for the degradation of triclosan (TCS). Figure 14A(i,ii) shows the fern-like morphology of BiVO_4_, while the composite shows well-dispersed N-biochar in intimate contact with BiVO_4_. Upon light irradiation (60 min), the BiVO_4_@N-Biochar catalysts showed 94.6% TCS degradation efficiency, which is much higher than pure BiVO_4_ (56.7%). As per LSMS and the *E. coli* (*Escherichia coli*) colony assessment studies, a detoxification efficiency of 72.3 ± 2.6% was determined, signifying a remarkable reduction in biotoxicity during photodegradation. The schematic representation of the possible charge transfer mechanism on BiVO_4_@N-Biochar is illustrated in Figure 14A(iii). Cao et al. [112] have reported the synthesis of Al-doped BiVO_4_ composites with the use of a simple hydrothermal method and evaluated for photocatalytic decomposition of MB dye (Figure 14B(i)). Different molar ratios of Bi to Al were used and calcined at a temperature of 500 °C under Ar gaseous environment (1h) to obtain Al-doped BiVO_4_. Based on optical spectroscopy, the band gap energies of pristine BiVO_4_, Al-0.03-BV, and Al-0.3-BV samples were determined to be 2.36, 2.40, and 2.41 eV, respectively. As shown in Figure 14B(ii), when isopropanol and potassium dichromate were used as scavengers, the degrading efficiency for MB was decreased, demonstrating that e^−^ and ^·^OH^−^ radicals are the active species in the degradation of dye molecules. Their experimental results confirm that optimal 30 mol% Al-BiVO_4_ (Al-0.3-BV) showed excellent photocatalytic activity for MB degradation due to the synergistic effect of appropriate Al doping and Al_2_O_3_ surface passivation. Based on transient photovoltage (TPV) and surface photocurrent (SPC) results, the coexistence of Al^3+^ and Al_2_O_3_ evolved, causing a synergistic effect for advancing e^−^ transfer and extending the lifetime of charge carriers. Al doping can result in transforming the surface morphology of BiVO_4_, as the polyhedron structure of BiVO_4_ becomes thinner and there is a reduced grain size (Figure 14B(iii–v)). Similarly, Wetchakun et al. [113] studied BiVO_4_/CeO_2_ nanocomposites through the co-precipitation and hydrothermal method. The different molar concentration of semiconductors’ constituent was fixed and evaluated for the degradation of dyes pollutants in water. The XRD results suggest that two different kinds of diffraction peaks confirmed the coexistence of mixed-phase, indicating BiVO_4_/CeO_2_ nanocomposite formation (Figure 14C(i)). Under light irradiation (>400 nm), the molar ratio 0.6:0.4 for BiVO_4_/CeO_2_ nanocomposite displayed the highest photocatalytic degradation activity for the removal of MB dye in water (Figure 14C(ii)).

#### 5.3.1. Activated Carbon, Carbon, and Other Adsorbents-Based Composites

Patil et al. [73] reported the synthesis of BiVO_4_/Ag/rGO hybrid architectures using a cost-effective hydrothermal method exhibiting impressive reaction rates for water oxidation and organic pollutant degradation reactions. Fern-like BiVO_4_ nanostructures were prepared and decorated on the surface of reduced graphene oxide (rGO) sheets (Figure 15A(i)), which can offer a large surface area and excellent electron transfer properties. They discovered that Ag nanoparticles can be reduced during hydrothermal reaction and deposited on the surface of BiVO_4_, forming a Schottky junction between Ag and BiVO_4_. Figure 15A(iii,iv)) depicts SEM and TEM images of nanocomposites, demonstrating different phases of Ag, rGO, and BiVO_4_. As shown in Figure 15A(ii), a complete degradation of MB dye within 120 min was achieved using a hybrid BiVO_4_/Ag 2%/rGO catalyst under simulated light irradiation, which is ~2.18 and ~1.25 times larger than pure BiVO_4_ and BiVO_4_/Ag photocatalysts. On the basis of PL and PEC results, the highest photocatalytic performance in this system is attributed to the combined effect of Ag and rGO, enabling the efficient promotion of e^−^/h^+^ separation across the interface and visible light absorption. The effect of graphene oxide on BiVO_4_ photocatalyst was also demonstrated by Zhang et al. by incorporating graphene oxide (GO) between the BiVO_4_ and NiOOH oxygen evolution catalysts (OEC). The results indicate that GO served as hole extraction layer due to its hole storage capability and improved the stability of the material. Meanwhile, GO employs the formation of p/n heterointerface with BiVO_4_ and encouraged the hole transfer from BiVO_4_ to NiOOH [114].

Graphitic carbon nitride (g-C_3_N_4_) is one of the best 2D semiconductor materials due to its large surface area and intriguing electronic properties. Incorporating g-C_3_N_4_ with BiVO_4_ can significantly modify the physicochemical properties and showed impressive photoactivity towards the degradation of organic pollutant dyes, CO_2_ reduction, and H_2_ generation reactions. Alhaddad et al. [115] reported a g-C_3_N_4_-incorporated Pt@BiVO_4_ nanocomposites catalyst for the detoxification of ciprofloxacin. A sol–gel reaction between Bi (NO_3_)_3_.5H_2_O and NH_4_VO_3_ was adopted using CH_3_COOH and HCl, forming BiVO_4_ nanoparticles. Then, using C_6_H_14_ as a solvent, solid dispersions of BiVO_4_ in different mass contents of 1.0, 2.0, 3.0, and 4.0 wt% and g-C_3_N_4_ were prepared, where it was agitated for 4h, resulting in the formation of heterostructures. At last, solid dispersions were then prepared via photoreduction. As shown in the TEM image (Figure 15B(i)), Pt nanoparticles can be photoproduced and deposited on the surface of BiVO_4_ and g-C_3_N_4_. A comparison study revealed that the 0.5 wt% Pt@4 wt% BiVO_4_-g-C_3_N_4_ heterojunction is optimal, displaying 5.0- and 3.7-times higher photocatalytic efficiency than pure g-C_3_N_4_ and BiVO_4_ for the decomposition of ciprofloxacin (Figure 15B(ii)). The plausible electron transfer mechanism over Pt@BiVO_4_-g-C_3_N_4_ during the photocatalytic removal of ciprofloxacin is illustrated in Figure 15B(iii). The creation of the p-n heterojunction can facilitate charge carrier separation, while Pt nanoparticles further assist in the increase in the light absorption and photocatalytic efficiency.

#### 5.3.2. Heterojunction Construction

Interestingly, monoclinic (m) BiVO_4_ and tetragonal zircon (tz) BiVO_4_ can form a heterojunction. For example, Dabodiya et al. [116] prepared mixed-phase BiVO_4_ (m:tz-60:40) using a microwave–hydrothermal method, which displays a 95% degradation efficiency for Rhodamine B dye. As shown in Figure 16A(ii), the effect of phase transition from tz-BiVO_4_ to m-BiVO_4_ was investigated in terms of its photocatalytic efficiency for the decomposition of RhB dye. On the basis of UV-reflectance and PL results, they discovered a reduction in the bandgap energy and facilitated e-/h^+^ separation at the m-BiVO_4_/tz-BiVO_4_ interfaces and enhanced photoactivity under visible light irradiation. Similarly, Patil et al. [117] reported the controlled synthesis of BiVO_4_ (pillars-like, dendrite-like, and microgranule-like) and the m-BiVO_4_/tz-BiVO_4_ heterojunction using simple a hydrothermal–solvothermal and solid-state reaction (Figure 16B(i)). The mixed-phase m-BiVO_4_/tz-BiVO_4_ heterojunction prepared through solvothermal reaction displayed the highest photodegradation efficiency of 95% for MB dye related to single-phase BiVO_4_ prepared through a hydrothermal (BVO-HDR; 79%) and solid-state reaction (BVO-SSR; 88%). Experimental results confirmed that temperature plays a critical role in the phase transformation of BiVO_4_. On the basis of PEC and EIS results, the high photocurrent density and reduction in internal resistance is confirmed, demonstrating that special dendritic architectures and the heterojunction effect is crucial to promote the e-/h^+^ separation and utilization. Figure 16B(ii) shows the possible electron transfer mechanism over the single-phase BiVO_4_ and m-BiVO_4_/tz-BiVO_4_ heterojunction for the photocatalytic degradation of organic dyes under light irradiation.

Recently, Li et al. [103] developed a ZnO/BiVO_4_ heterojunction thin films catalyst using a simple chemical bath deposition and electrodeposition method. The as-formed three-dimension choral-like ZnO/BiVO_4_ displayed an excellent photoelectrocatalytic tetracycline degradation efficiency of 84.5%. Figure 17A(i,ii) shows the schematic representation of the BiVO_4_/ZnO electrode and charge transfer process during photoelectrocatalysis. According to the radical scavenger’s test, •O_2_^−^ and •OH were found to be major active species responsible for tetracycline degradation. As shown in Figure 17A(iii), ZnO/BiVO_4_ showed a tetracycline degradation efficiency 84.5%.

Yan et al. [118] fabricated the BiVO_4_/Ag_3_VO_4_ heterojunction using a simple hydrothermal and coprecipitation method and evaluated its photocatalytic performance towards RhB dye degradation. Figure 17B(i) shows the TEM image of hybrid BiVO_4_/Ag_3_VO_4_ and two sets of lattice fringes, indicating an intimate interface between the two semiconductor and heterojunction creation. As shown in Figure 17(Bii), the difference in the molar ratio showed varied photocatalytic activity. Among them, a 10:1 mol ratio of the BiVO_4_:Ag_3_VO_4_ heterojunction displayed the highest 95% degradation efficiency for RhB dye, which is 10- and 3.4-times higher than pure BiVO_4_ and Ag_3_VO_4_, respectively. The schematic representation of the photocatalytic reaction mechanism is illustrated in Figure 17B(iii). On the basis of electrochemical impedance analysis (Figure 17B(iv)), a small semicircle for BiVO_4_:Ag_3_VO_4_ suggests a lowest internal resistance and faster electron transfer process due to the heterojunction effect and the effective separation of the photo-induced charges carriers.

Bao et al. [119] designed a facet–heterojunction Z-scheme photocatalyst AgBr-Ag-BiVO_4_ {010} to increase the photoactivity of BiVO_4_ for the inactivation of pathogenic bacteria and the degradation of organic dyes from wastewater. First, facet-controlled BiVO_4_ {010} was obtained using the facile hydrothermal method, and then Ag nanoparticles were deposited through photoreduction, while in situ chemical treatment in the presence of KBr and Fe(NO_3_)_3_ enabled the transformation of the outermost layer of Ag nanoparticles to AgBr. Interestingly, Ag nanoparticles can be selectively deposited on the BiVO_4_ {010} facets and transform. Figure 18i shows the FESEM image of the AgBr-Ag-BiVO_4_ {010} heterojunction and highly dispersed AgBr nanoparticles, which showed an increase in particle sizes upon transformation. AgBr-Ag-BiVO_4_ {010} displayed the highest photocatalytic inactivation for Escherichia coliK-12, which is ∼4 and 15 times the reference Ag-BiVO_4_ {010} and BiVO_4_, respectively (Figure 18iii). The possible charge transfer mechanism during the photocatalytic inactivation of bacterium is illustrated in Figure 18ii. The photoluminescence (PL) spectroscopy and PEC results suggested a suppression in the charge recombination upon the heterojunction, and the electron paramagnetic resonance (EPR) results revealed that h^+^, ^•^OH, and ^•^O_2_^−^, are the major active species for the degradation of RhB dyes in wastewater.

#### 5.3.3. BiVO_4_-Based S-Scheme and Z-Scheme Nanocomposite Materials

As mentioned above, it is difficult for single BiVO_4_ semiconducting materials to offer a strong visible light response and high redox capability simultaneously. So, the coupling of BiVO_4_ with another suitable semiconducting material to synthesize S-scheme- and Z-scheme-type composite materials has received significant attention. The band structure of both the S-scheme- and Z-scheme-type composite materials is shown in Figure 19.

In both composite materials, the high CB semiconductor materials combine with high VB semiconducting materials, whereas they follow two different pathways for photo-generated electron–hole pairs separation and transfer. In Z-scheme-type materials, the photo-generated electron–hole pairs are separated by an internal electric field between the two semiconductor interfaces, whereas in the case of S-scheme-type materials, the charge carrier separation and transfer occur through an internal electric field, energy band bending, and Coulomb gravity. Furthermore, there are two types of Z-scheme-type materials, such as direct Z-scheme-type materials and mediator (indirect)-based Z-scheme-type materials. In indirect Z-scheme-type materials, metals and non-metals, carbon materials, and quantum dots, etc., are used as mediators. For example, Li et al. coupled high-valence-band-edge BiVO_4_ with high-conduction-band-edge g-C_3_N_4_ material through a wet impregnation-calcination method which yielded Z-scheme BiVO_4_/g-C_3_N_4_ materials. The calculated CB and VB potentials of g-C_3_N_4_ and BiVO_4_ are 1.20 and 1.54 eV and 0.46 and 2.86 eV, respectively. Under visible light irradiation, the electron–hole pairs are generated in both the semiconductors. Subsequently, the electrons present in the CB of BiVO_4_ combines with holes present in the VB of g-C_3_N_4_, so the electrons in the CB of g-C_3_N_4_ and holes in the VB of BiVO_4_ are efficiently separated and possess a higher potential than the generation potential of the reactive radicals (^•^OH and O_2_^•–^). Consequently, a higher concentration of reactive radicals is generated, which showed higher photocatalytic activity in the degradation of malachite green (MG) dye in the presence of visible light irradiation and H_2_O_2_ [120]. Similarly, Hu et al. developed a g-C_3_N_4_/BiVO_4_-based S-scheme system using hydrothermal methods for the degradation of paraben preservative in the presence of visible light and natural solar light irradiation. The CB and CB of g-C_3_N_4_ were located at–1.3 and + 1.44 V vs. RHE at pH 0, respectively, and the CB and VB of BiVO_4_ were sited at + 0.09 and + 2.4 V vs. RHE, respectively. The energy difference present in the mixed g-C_3_N_4_/BiVO_4_ system would allow the transfer of electrons of g-C_3_N_4_ to BiVO_4_, which leads to the positively charged region on the g-C_3_N_4_ side and negatively charged region on the BiVO_4_ side. Therefore, there is a generation of an inner electric field at the interface of g-C_3_N_4_/BiVO_4_, with the direction from BiVO_4_ to g-C_3_N_4_. Under irradiation, the photoexcited electrons on the CB of BiVO_4_ combined with the holes on the VB of g-C_3_N_4_ lead to the efficient separation of electrons and holes on the CB of g-C_3_N_4_ and on the VB of BiVO_4_ for the production of reactive radicals, which could participate in the degradation of paraben [121]. Similarly, various researchers have developed BiVO_4_-based Z-scheme and S-scheme materials for the degradation of dyes and other compounds; however, this has been scarcely studied for the application of VOC degradation.

## 6. Volatile Organic Compounds Degradation Application

As described in the above sections, volatile organic compounds (VOCs) pose a serious threat to environment and human health. VOCs are mainly BTEX (benzene, toluene, ethylbenzene, xylene), acetylene, acetone, ethylene, trichloroethylene, benzaldehyde, acetaldehyde, isopropanol, hexane, etc., and are mainly released by human activities through outdoor and indoor sources. VOCs create significant health issues; they specifically cause allergies, cancer, and they slow down and damage the nervous and respiratory system. Therefore, significant research activities are put forward for controlling and degradation of VOCs. Among those, the photocatalytic oxidation/degradation of VOCs has received great attention because of the simple operation and reaction conditions, low cost, and the fact that it completely degrades VOCs, and renewable solar energy can be used for their degradation. Among photocatalysts, BiVO_4_ is a better visible-light-responsive system for VOCs degradation, which are described in this section and Table 3.

Hu et. al. synthesized a BiVO_4_/TiO_2_ heterojunction photocatalyst using the sol–gel method and evaluated the photocatalytic oxidation of gaseous benzene under UV light and simulated solar light irradiation. The BiVO_4_ with a loading percentage of 0.5% presented a higher photocatalytic oxidation of benzene (66.8% conversion, Figure 20a) and a high amount of CO_2_ production (Figure 20b) compared to the other percentage loaded materials and bare TiO_2_ and BiVO_4_ materials. The improved visible light absorption by the introduction of BiVO_4_ and the activated species (•OH) are responsible for the high activity of the BiVO_4_/TiO_2_ heterojunction photocatalyst [123]. Furthermore, Zhao et al. developed CuO/BiVO_4_ hollow nanospheres using the sol–gel method followed by the impregnation method and demonstrated visible light photocatalytic activity by the degradation of gaseous toluene. The results revealed that 5% CuO-loaded BiVO_4_ hollow nanospheres showed higher visible light photocatalytic activity (85%) in toluene degradation compared to other percentage-loaded composites and bare materials [124]. Chen et al. hydrothermally synthesized BiVO_4_/α-Fe_2_O_3_ composites and calcined them at various temperatures (250, 350, 450, and 550 °C) and evaluated their efficiency by the degradation of benzene under UV light irradiation. The composites calcined at 350 °C exhibited higher benzene removal efficiency (66.87%) than other composites and bare α-FeOOH, Fe_2_O_3_, and BiVO_4_ materials [131]. However, still the binary composites showed low degradation efficiency due to the low separation and transfer rate of photogenerated charge carriers. In order to further enhance the photocatalytic efficiency of BiVO_4_, BiVO_4-_based Z-scheme and ternary nanocomposites were prepared, and also metal nanoparticles were introduced into BiVO_4_. For example, a coral-like direct Z-scheme BiVO_4_/g-C_3_N_4_ was synthesized by Sun et al. for the degradation of toluene under visible light irradiation. The visible light absorption of BiVO_4_ was significantly increased after g-C_3_N_4_ loading and also promoted the separation of photogenerated electron hole pairs. The improved separation of photogenerated electron hole pairs on the direct Z-scheme BiVO_4_/g-C_3_N_4_ material led to a higher toluene degradation efficiency compared to bare materials [130]. Furthermore, Li et al. studied automobile exhaust gas purification by improving the adsorption capacity of volatile compounds (automobile exhaust gases HC, NO, and CO) onto g-C_3_N_4_/BiVO_4_ composites through introducing tourmaline powder into g-C_3_N_4_/BiVO_4_ composites. The introduction of tourmaline powder considerably increases the adsorption capacity of the automobile exhaust gas molecules by releasing negative ions, which enhances the contact between the automobile exhaust gas molecules and the g-C_3_N_4_/BiVO_4_ composite material. The enhancement in the adsorption capacity improves the hydrocarbon, CO, and NO purification efficiency by 1.73, 1.74, and 2.52 times compared to pure g-C_3_N_4_ [134]. In addition, the charge carrier’s separation and transport efficiency was enhanced by introducing an oxygen vacancy (OVs) on BiVO_4_ using the electrochemical reduction process_._ The ternary nanocomposite (BiVO_4_/WO_3_/TiO_2_) was prepared by coupling OVs-BiVO_4_ with WO_3_/TiO_2_ nanotubes for toluene gas degradation. The OVs-BiVO_4_/WO_3_/TNTs displayed a 28-times higher photocurrent intensity compared to pristine BiVO_4_/WO_3_/TNTs, which leads to higher photocatalytic toluene gas degradation. Furthermore, the stability of the composite materials was also enhanced by introducing OVs [135]. Recently, Zhu et al. developed triangular Ag nanoplates (AgNPs)-loaded BiVO_4_ for the degradation of gaseous formaldehyde (HCHO) under the irradiation of daylight lamp as a visible light source. The loading of triangular Ag nanoplates significantly decreases the recombination rate of photogenerated electron–hole pairs that extends the lifetime of charge carriers, which leads to a high HCHO oxidation efficiency. The plasmonic effect of AgNPs was also the reason for the enhancement of the catalytic activity [137]. Similarly, Shi et al. prepared activated carbon from semi-coke waste generated during the processing of coal and was loaded into a ternary BiVO_4_–BiPO_4_–g-C_3_N_4_ Z-scheme heterojunction photocatalyst using a one-step sol–gel method for the degradation of toluene under visible light irradiation. The activated carbon-loaded ternary composites showed a 2.43-times higher photocatalytic activity (85.6%) than the pure photocatalyst in the degradation of toluene under 60% relative humidity and 0.5 g/cm^3^ of composite material. The enhanced adsorption of toluene by activated carbon loading and the improved visible light response leads to higher activity in toluene VOC degradation [139]. Likewise, the development of BiVO_4_-based visible light active materials for VOC decomposition is still growing; however, there are some limitations that affect the possible utilization of BiVO_4_ on a commercial scale, which are described in the forthcoming sections.

## 7. Summary and Outlook

BiVO_4_ has been identified as one of the most promising visible-light-responsive photocatalytic materials (low bandgap energy, ~2.4 eV) for the degradation of various pollutant molecules, including VOCs. However, the materials are facing significant issues such as the high recombination rate of photogenerated charge carriers, the inappropriate position of CB of BiVO_4_, and the low redox ability of charge carriers. So, various engineering modifications have been performed to improve these limitations, which leads to high catalytic efficiency. Therefore, in summary, we have reviewed the synthesis methodologies of BiVO_4_ and various engineering modifications of BiVO_4_, such as changes in the morphology, metal and non-metal loading, heterojunction formation, Z-scheme- and S-scheme-type materials development and support on high-surface-area adsorbents, etc., followed by their application in VOCs degradation, which was reviewed in detail in this paper, for which it showed a greater performance.

Though the modified BiVO_4_ material showed an efficient visible light photocatalytic performance, it is still lagging behind in terms of commercial applications because of its poor reusability and low lifetime. Furthermore, for the better commercial utilization and recovery of materials, there is a need for an immobilized photocatalytic reactor. However, the presently available BiVO_4_-based immobilized reactor systems have a low mass transfer effect (both external and internal). In addition, morphologies-controlled synthesis generally results in a relatively larger size of BiVO_4_ material, which is easier to agglomerate and it decreases the active surface sites, thereby reducing the catalytic performance. Hence, future research needs to focus on advanced synthesis techniques, such as atomic layer deposition methods, plasma treatment, and other micro techniques, to achieve a greater precision in the control of the morphology and the production of an appropriate size of the BiVO_4_. This should be followed by smaller size BiVO_4_-based materials as these could be effectively used to make an efficient photocatalytic surface with a high mass-transfer effect. Future research also needs to be focused on the production of concrete evidence for the photogenerated separation of electron–hole pairs and transfer pathways/mechanism in heterojunction and Z-scheme- and S-scheme-type materials. Finally, the toxicity assessment of the synthesized materials as well as degraded solution requires increased attention in the near future for the commercial utilization of a developed photocatalytic system. We hope that this review article has created a solid foundation for the development of BiVO_4_-based composite materials with a high performance, as well as their associated technologies, for the decomposition of VOCs.

## Figures and Tables

**Figure 1 nanomaterials-13-01528-f001:**
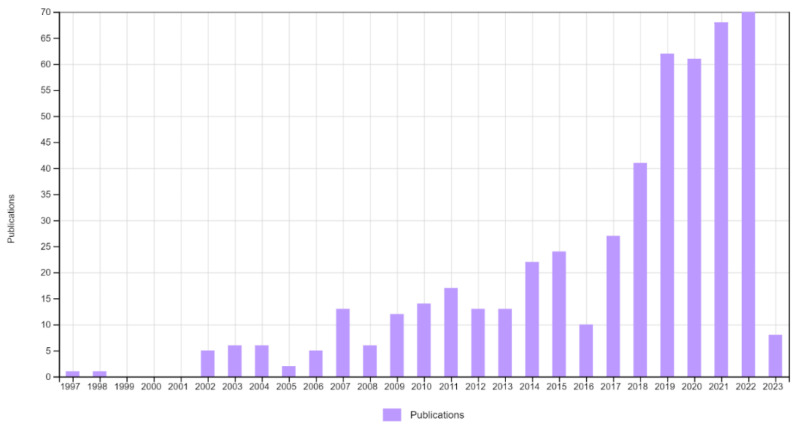
Number of publications on photocatalytic degradation of volatile organic compounds. (Source: Web of Science, searched keywords: photocatalytic oxidation of volatile organic compounds, photocatalytic degradation of volatile organic compounds, nitrogen-doped TiO_2_, and volatile organic compounds (VOCs)).

**Figure 2 nanomaterials-13-01528-f002:**
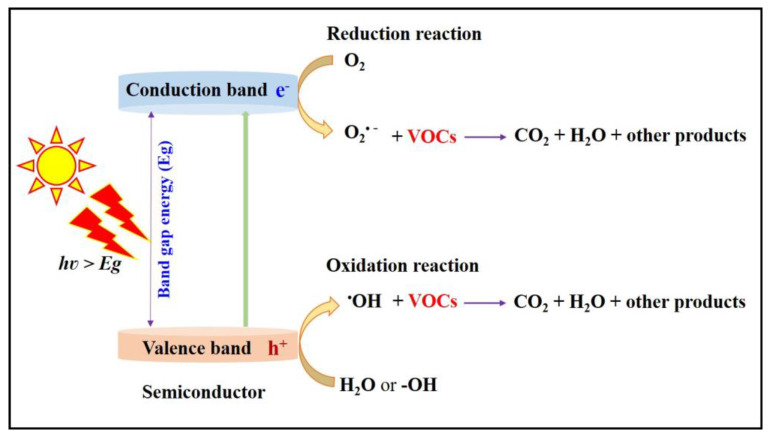
Schematic representation of mechanism of photocatalytic oxidation.

**Figure 3 nanomaterials-13-01528-f003:**
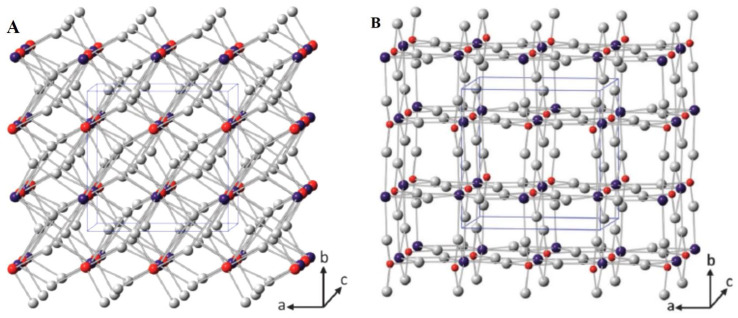
(**A**) The crystal structures of (S) tetragonal scheelite (red: V, purple: Bi, and gray: O) and (**B**) zircon-type BiVO_4_ (red: V, purple: Bi, and gray: O) (reprinted with permission from Ref. [49]; 2013, copyright from Elsevier).

**Figure 4 nanomaterials-13-01528-f004:**
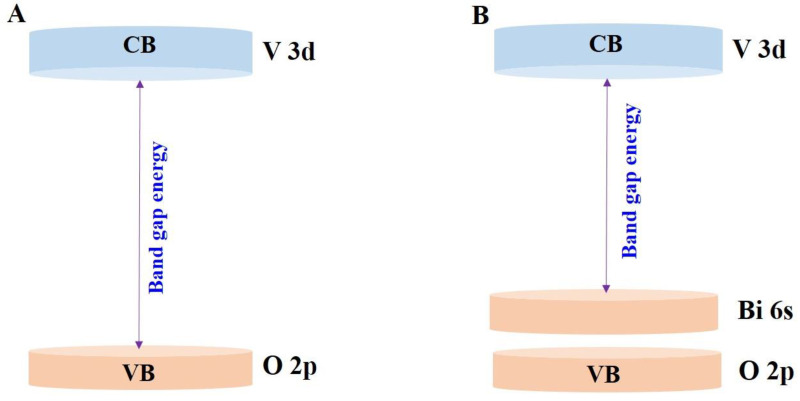
(**A**) Band structures of the tetragonal zircon-type BiVO_4_ and (**B**) band structure of monoclinic scheelite-type BiVO_4_ (reprinted with permission from Ref. [43]; 1999, copyright from American Chemical Society).

**Figure 5 nanomaterials-13-01528-f005:**
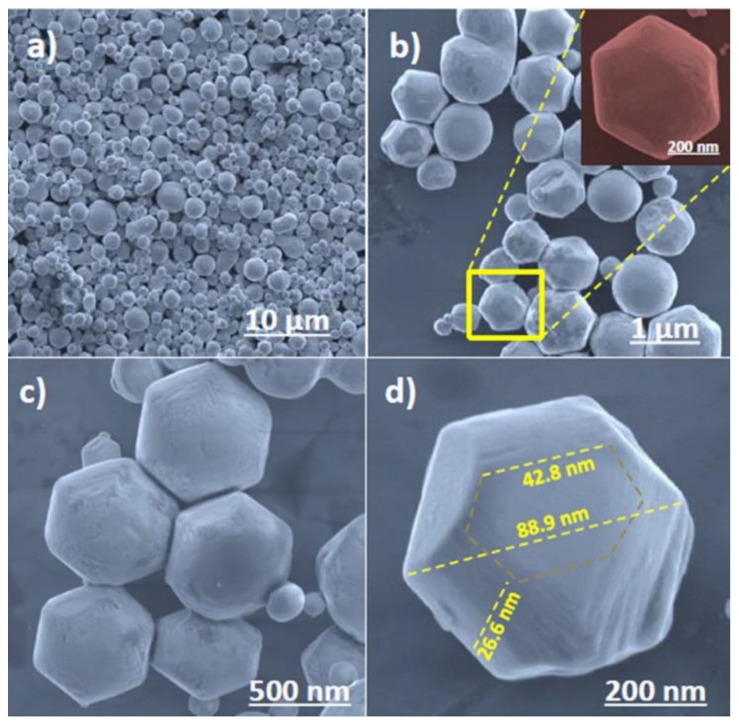
FE-SEM images of pure m-BiVO_4_ photocatalyst (**a**) at 10 µm scale bar, (**b**) at 1 µm scale bar. Inset at 200 µm scale bar showing the hexagonal structure of a single NP, (**c**) at 500 µm scale bar showing uniformly neighboring NPs, and (**d**) at 200 µm scale bar showing the rough diameter and thickness of a single NP. (Reprinted with permission from [42]; copyright from Nature).

**Figure 6 nanomaterials-13-01528-f006:**
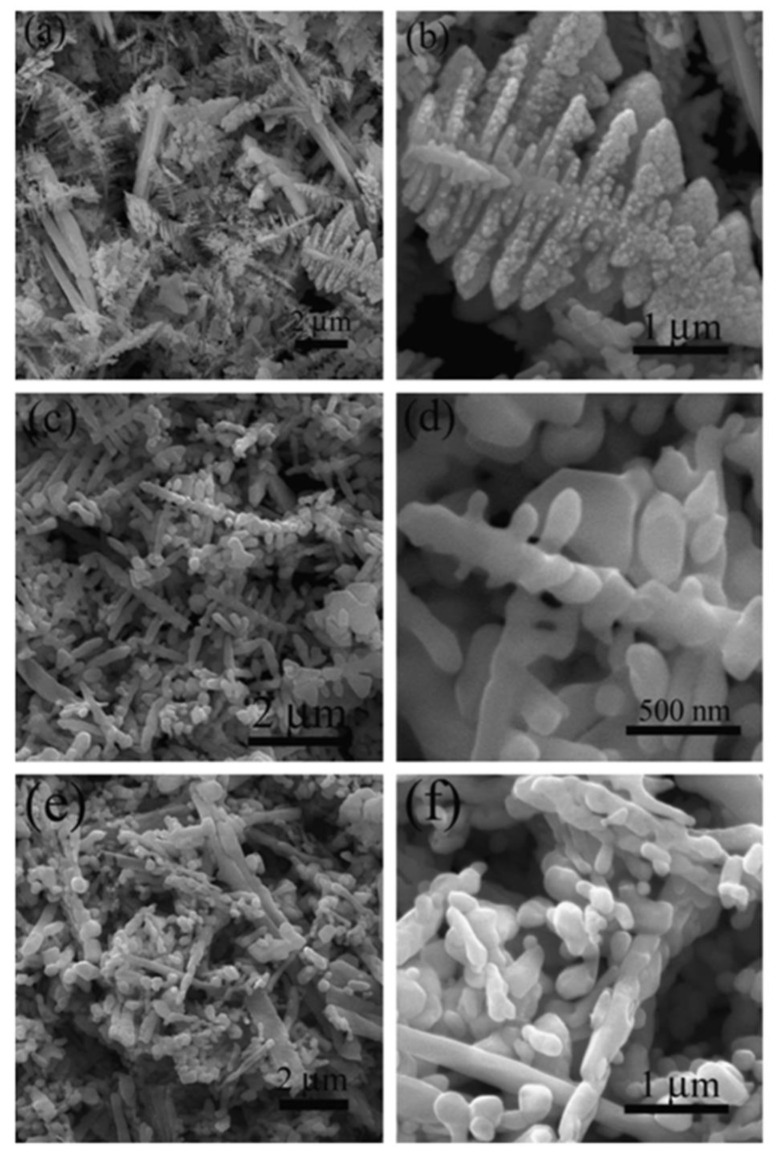
FE-SEM images of the BiVO_4_ samples attained from (**a**,**b**) 100 °C, (**c**,**d**) 140 °C, and (**e**,**f**) 180 °C hydrothermal temperatures. (Adopted with permission from [52]; copyright from Elsevier.

**Figure 7 nanomaterials-13-01528-f007:**
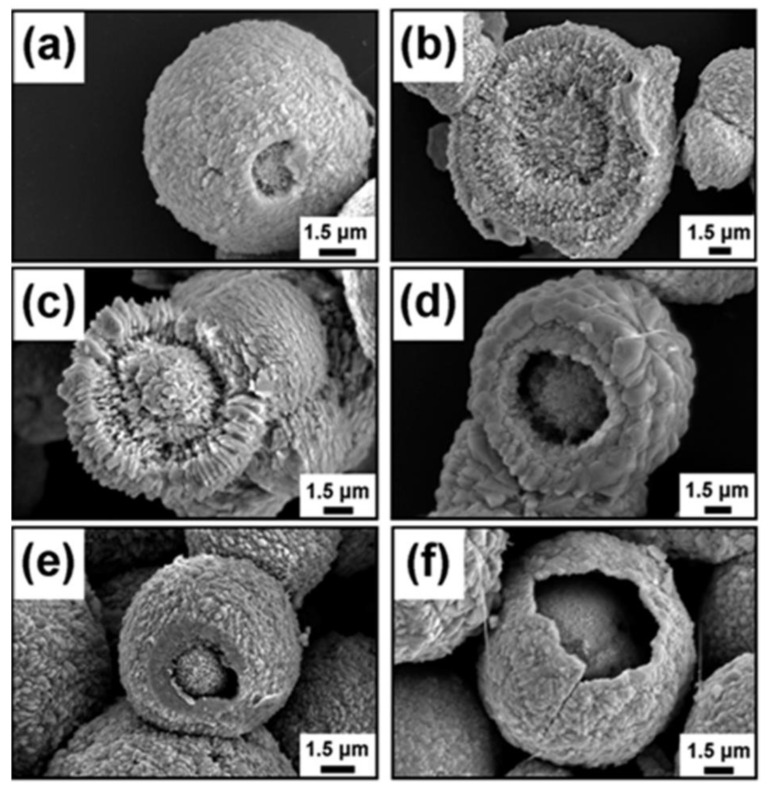
Close-up sequence of the evolution of the CSS BiVO_4_ hollow spheres: (**a**) 1 h, (**b**) 3 h, (**c**) 5 h, (**d**) 8 h, (**e**) 10 h, and (**f**) 15 h. (Reprinted with permission from [55]; copyright permission from the Royal Society of Chemistry).

**Figure 8 nanomaterials-13-01528-f008:**
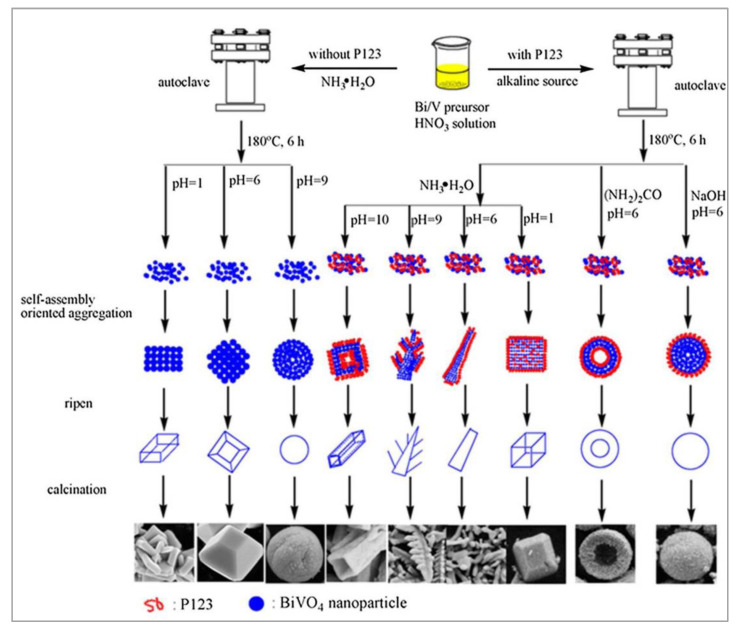
Schematic diagram of growth mechanisms of the BiVO_4_ particles with numerous morphologies under dissimilar hydrothermal conditions. (Reprinted with permission from [56]; copyright from Elsevier).

**Figure 9 nanomaterials-13-01528-f009:**
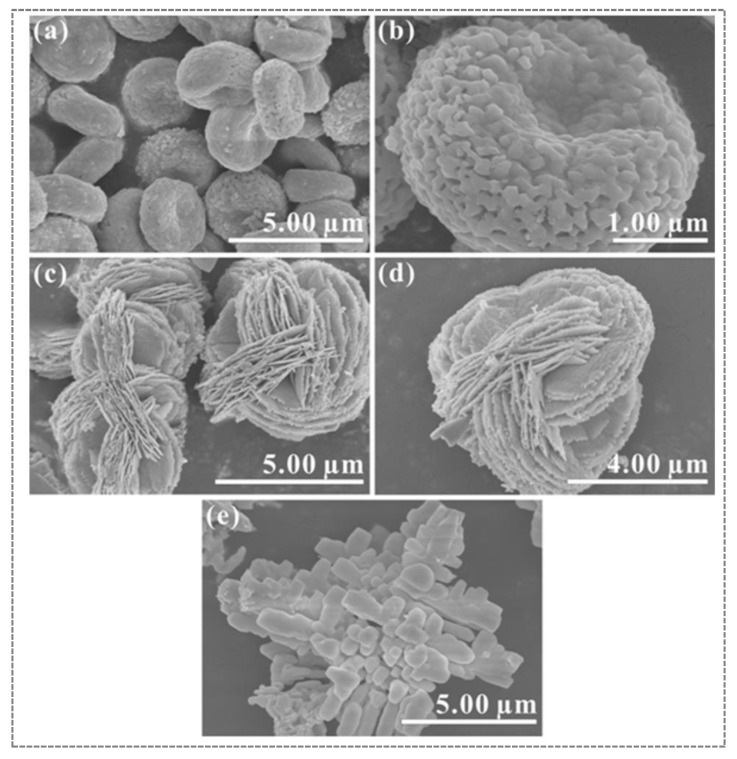
FE-SEM images of (**a**,**b**) S-BiVO_4_, (**c**,**d**) A-BiVO_4_, and (**e**) N-BiVO_4_. (Reprinted with permission from [62]; copyright from Elsevier).

**Figure 10 nanomaterials-13-01528-f010:**
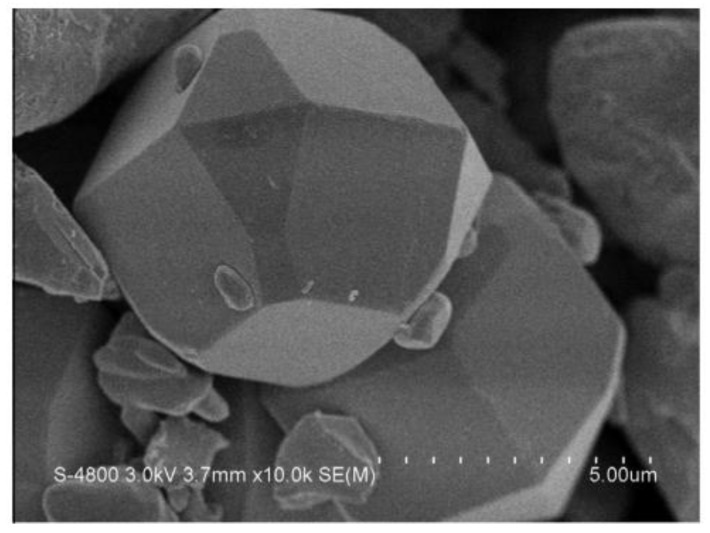
SEM image of the as-prepared BiVO_4_ photocatalyst. (Reprinted with permission from [64]; copyright from Elsevier).

**Figure 13 nanomaterials-13-01528-f013:**
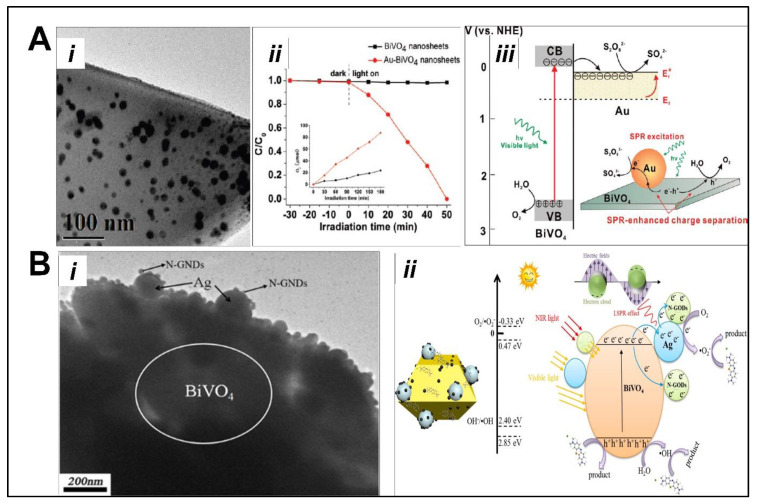
(**A**) Au-doped BiVO_4_ heterogeneous nanostructures. (**i**) TEM image of Au−BiVO_4_ nanoplates. (**ii**) Photocatalytic degradation performance of MO dye with Au−BiVO_4_ under light irradiation. (**iii**) Schematic representation of mechanism of photocatalytic water oxidation using Au−BiVO_4_ using sacrificial agent S_2_O_8_^2−^. Facile transfer of electron from BiVO_4_ to Au shifts fermi energy level (E_f_) of Au to more negative potentials. Reproduced with permission from Ref. [97]. (**B**) N−GNDs/Ag/BiVO_4_ photocatalyst for degradation of TC•HCl. (**i**) HRTEM image of N−GNDs/Ag/BiVO_4_. (**ii**) Schematic illustration for electron transfer mechanism for degradation of TC•HCl. Reproduced with permission from Ref. [99].

**Figure 14 nanomaterials-13-01528-f014:**
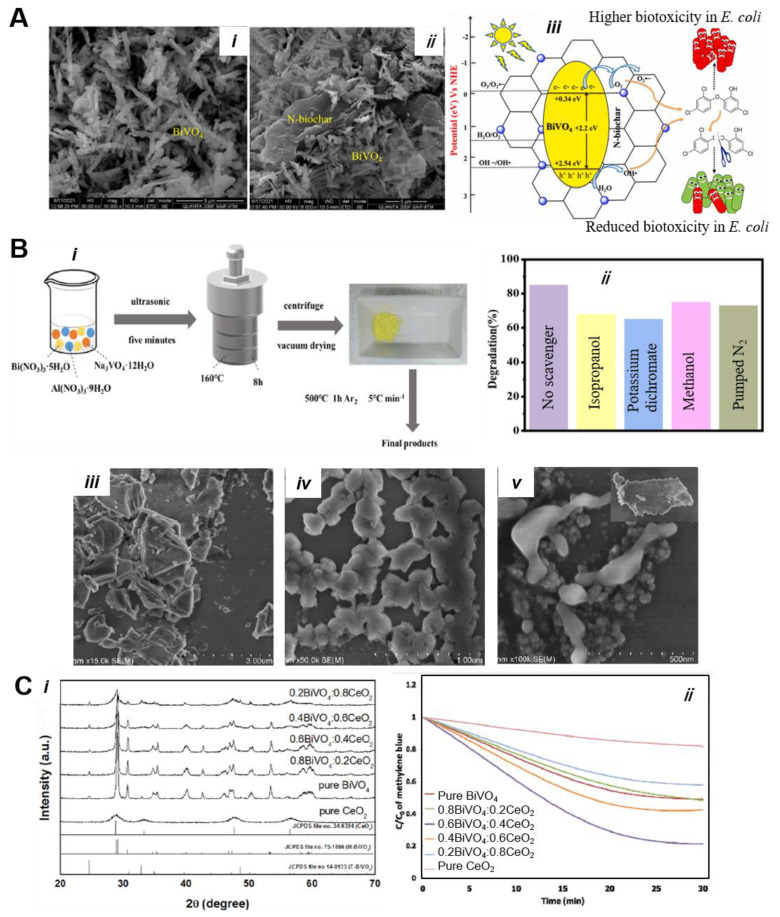
(**A**) BiVO_4_@N-Biochar nanocomposite photocatalyst for detoxification of triclosan. (**i**) SEM image of BiVO_4_ and (**ii**) SEM image of BiVO_4_@N-Biochar. (**iii**) Schematics for degradation mechanism of TCS using BiVO_4_@N-biochar under light irradiation. (Reprinted with permission from Ref. [111]). (**B**) Al-Doped BiVO_4_ composites, (**i**) schematic representation of hydrothermal assisted synthesis, and (**ii**) comparison of photodegrading efficiency of Al-BiVO_4_ composites in presence of different scavengers. SEM image of (**iii**) pure BiVO_4_ (**iv**) 0.03-BV and (**v**) 0.3-BV composite photocatalysts. Reprinted with permission from Ref. [112]. (**C**) BiVO_4_/CeO_2_ nanocomposites as visible light photocatalysts, (**i**) XRD patterns of pure BiVO_4_, pure CeO_2_, and BiVO_4_/CeO_2_ nanocomposites with various mole ratios, and (**ii**) comparison of photoactivity of different photocatalysts for MB degradation. Reprinted with permission from Ref. [113].

**Figure 15 nanomaterials-13-01528-f015:**
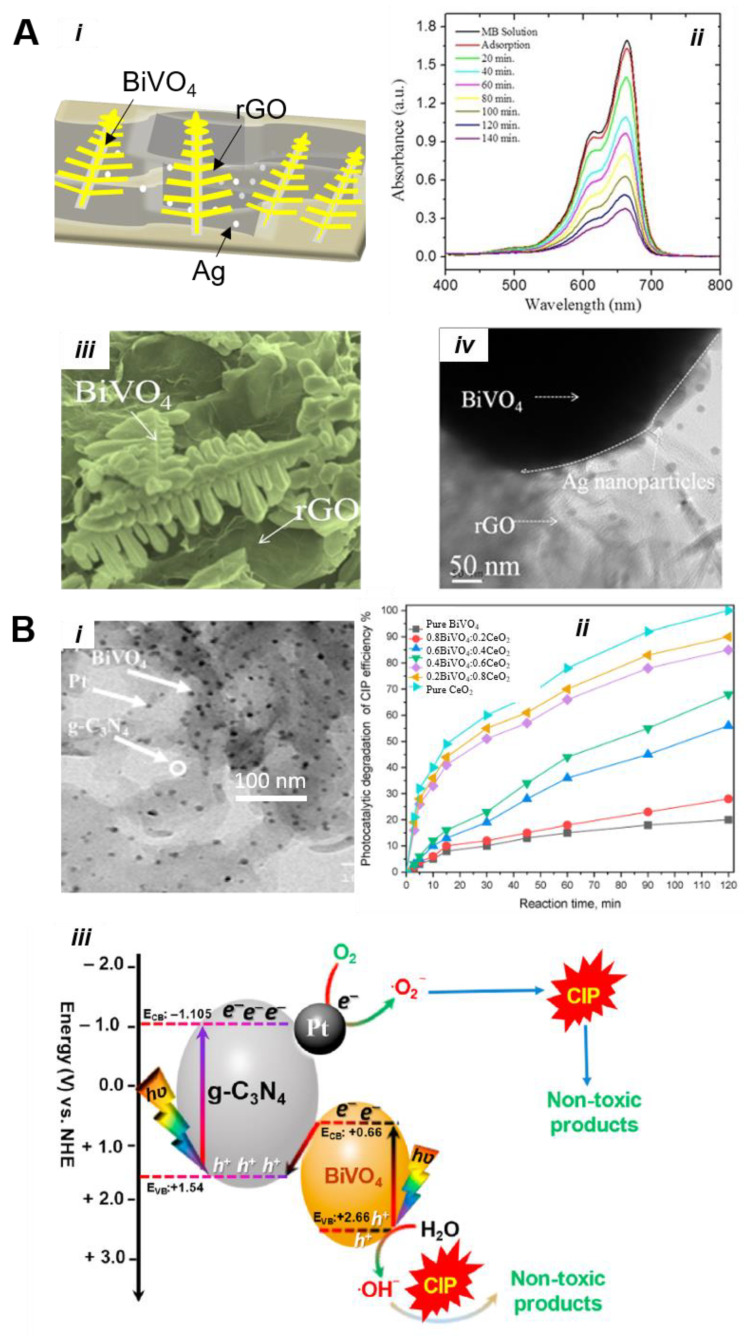
(**A**) BiVO_4_/Ag/rGO nanocomposite architectures. (**i**) Schematic illustration of growth of BiVO_4_ and Ag Nps on rGO nanosheets, (**ii**) photocatalytic degradation performance of BiVO_4_/Ag/rGO for MB dye under light, (**iii**) SEM image of BiVO_4_/Ag/rGO, and (**iv**) HRTEM image showing the three components rGO, Ag, and rGO. Reprinted with permission from Ref. [73]. (**B**) (**i**) TEM image of Pt@BiVO_4_-g-C_3_N_4_ nanocomposite, (**ii**) degradation ratio of ciprofloxacin using composite photocatalysts, and (**iii**) photocatalytic electron transfer mechanism. Reprinted with permission from Ref. [115].

**Figure 16 nanomaterials-13-01528-f016:**
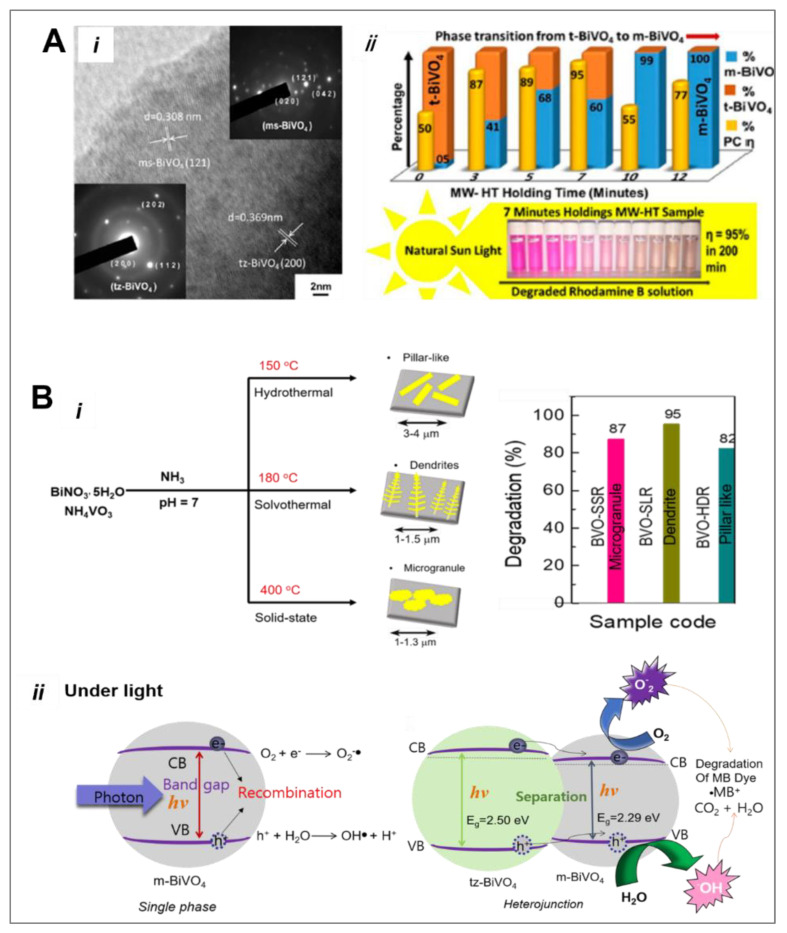
(**A**) Heterojunctions of BiVO_4_ emerged from mixed of tetragonal and monoclinic crystalline phase. (**i**) HRTEM image of BiVO_4_ indicating the mixed-phase and (**ii**) schematic showing the effect of phase transition on the photocatalytic degradation for RhB dye. Reprinted with permission from Ref. [116]. (**B**) (**i**) Controlled synthesis of BiVO_4_, heterojunctions, and photoactivity thereof. (**ii**) Photocatalytic dye degradation mechanism showing electron transfer in single-phase BVO and tz-BVO/m-BVO heterojunction under light illumination. Reprinted with permission from Ref. [117].

**Figure 17 nanomaterials-13-01528-f017:**
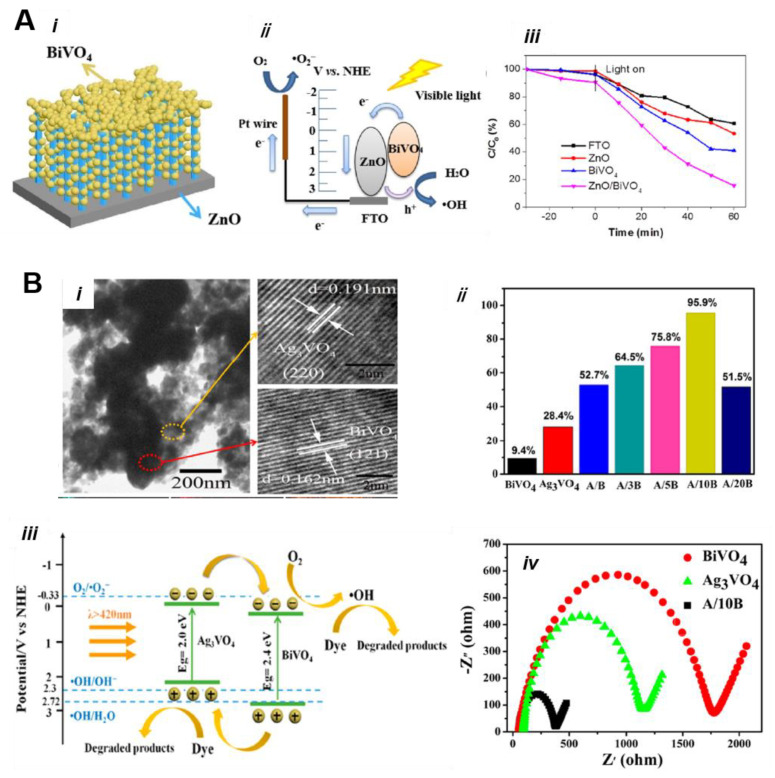
(**A**) BiVO_4_/ZnO heterojunction. (**i**) Schematic representation of coral-like BVO/ZnO nanostructures, (**ii**) schematic for possible electron transfer mechanism in BiVO_4_/ZnO junction catalyst, and (**iii**) photoelectrocatalytic removal efficiency of aqueous tetracycline (50 mL, 20 mg/L, 1.2 V vs. Ag/AgCl) in the presence of varied samples. Reprinted with permission from Ref. [103]. (**B**) Visible-light-driven BiVO_4_/Ag_3_VO_4_ heterojunction, (**i**) TEM and HRTEM image of BiVO_4_/Ag_3_VO_4_ heterojunction, and (**ii**) comparison of photocatalytic RhB dye using different catalysts. (**iii**) Photocatalytic reaction mechanism and (**iv**) comparison of electron impedance spectroscopy (EIS) results. Reprinted with permission from Ref. [118].

**Figure 18 nanomaterials-13-01528-f018:**
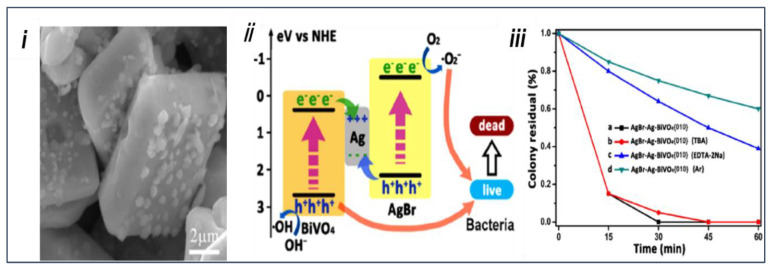
Z-Scheme BiVO_4_{010} microplates deposited with AgBr−Ag nanoparticles as photocatalyst. (**i**) SEM image of AgBrAg−BiVO_4_ {010} and (**ii**) schematics for mechanism of bacterial inactivation using AgBr−Ag−BiVO_4_ {010} under light irradiation. (**iii**) Photocatalytic inactivation for Escherichia coliK-12 (reprinted with permission from Ref. [119].

**Figure 19 nanomaterials-13-01528-f019:**
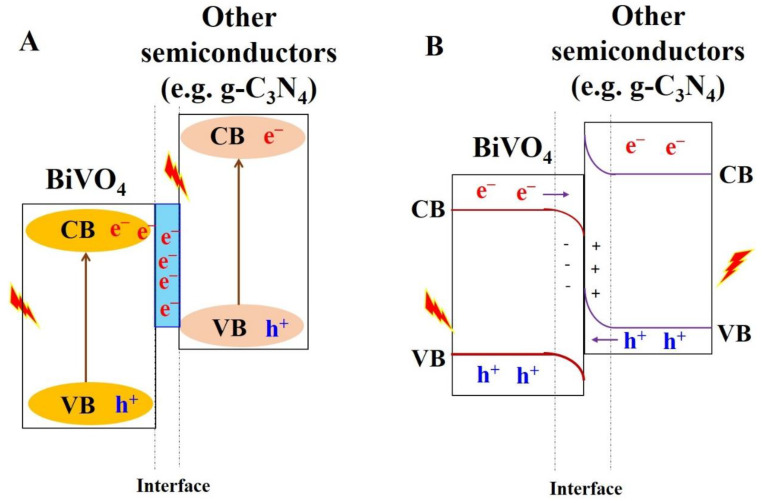
The band structure of (**A**) Z-scheme- and (**B**) S-scheme-type composite materials.

**Figure 20 nanomaterials-13-01528-f020:**
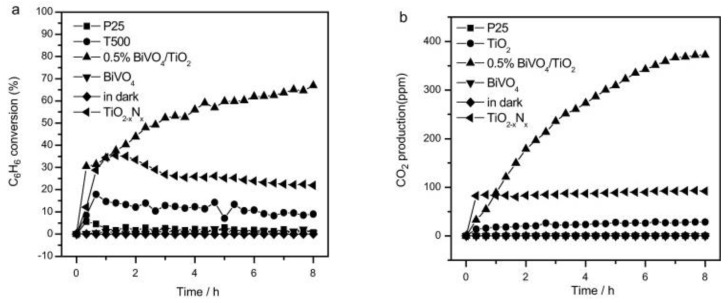
Photocatalytic oxidation of benzene (**a**) and the amount of CO_2_ production (**b**) using BiVO_4_/TiO_2_ and bare nanomaterials under visible light irradiation (reprinted with permission from [123]; 2011, copyright from Elsevier.

**Table 1 nanomaterials-13-01528-t001:** Basic semiconductor materials used for photocatalytic applications.

S. No	Semiconductor	Bandgap Energy (Eg), eV	VB Position	CB Position
1.	TiO_2_	3.2	+3.1	−0.1
2.	ZnO	3.2	+3.0	−0.2
3.	WO_3_	2.8	+3.0	+0.4
4.	ZrO_2_	5.2	+5.0	+0.2
5.	SnO_2_	3.8	+4.1	+0.3
6.	SrTiO_3_	3.2	+3.1	−0.1
7.	CuO	2.1	+2.36	+0.26
8.	ZnS	3.7	+1.4	−2.3
9.	CdS	2.5	+2.1	−0.4
10.	BiVO_4_	2.3–2.4	+2.7	+0.3
11.	Bi_2_MoO_6_	2.63	+2.9	+0.27
12.	Bi_2_WO_6_	2.77	+3.05	+0.28
13.	BiOF	3.64	+4.24	+0.6
14.	BiOCl	3.22	+3.4	0.18
15.	BiOBr	2.64	+3.0	+0.36
16.	BiOI	1.77	+2.32	+0.55
17.	Bi_2_O_3_	2.1–2.8	+2.48	+0.38

**Table 2 nanomaterials-13-01528-t002:** Synthesis methodologies, morphology, the photocatalytic activity of pollutant(s) and degradation efficiency of BiVO_4_-based materials.

Method of Synthesis	Morphology	Photocatalytic Activity of Pollutant(s) and Degradation Efficiency	Refs.
Surfactant- and template-free hydrothermal method	Truncated square, 18-sided	Pollutant: MB Dye (20 ppm) Light source: 1000 W xenon lamp % degradation: 91% after 60 min	[42]
Hydrothermal method using EDTA as a chelating agent	2D star-like crystals	Pollutant: MB Dye (15 ppm) Light source: 500-W xenon lamp % degradation: 99.3% after 25 min	[51]
Additive-free hydrothermal method	Dendritic structure of BiVO_4_	Pollutant: RhB Dye (10 ppm) Light source: 500 W xenon lamp % degradation: 91% after 210 min	[52]
Template-free hydrothermal method	Olive-like BiVO_4_	Pollutant: MB Dye (10 μM) Light source: 300 W xenon lamp % degradation: 84.1–95.7% (Different pH value) after 180 min	[53]
Surfactant-free hydrothermal method	Octahedral	Pollutant: MB Dye (10 ppm) Light source: low power xenon lamp. % degradation: 50–60% after 120 min	[54]
Surfactant- and template-free hydrothermal method	Plate morphology and biscuit morphology	Pollutant: RhB Dye (10^−5^ mol/L) Light source: 500 W xenon lamp % degradation: 99% after 270 min	[55]
Hydrothermal method in the presence of triblock copolymer P123 as a surfactant	Polyhedral, rod-like, tubular, leaf-like, and spherical	Pollutant: MB Dye 1.0 × 10^−5^ mol/L) Light source: 300 W xenon lamp % degradation: 90% after 120 min	[56]
Hydrothermal route using of SDBS as an anionic surfactant	2D single-crystal nanosheets	Pollutant: RhB Dye (2.09 10^−4^ mol dm^−3^) Light source: Sunlight % degradation: - 95% after 100 min	[57]
Hydrothermal process	Fibrous or needle-like sepiolite distributed peanut-shape monoclinic BiVO_4_ surface	Pollutant: TCs (5 ppm); MB Dye (10 ppm) Light source: LED lamp % degradation: TCs 78%; MB Dye 96% after 240 min	[58]
CTAB-assisted hydrothermal method	Snow-like	Pollutant: CIP (10 ppm) Light source: 500 W xenon lamp % degradation: 98.5% after 70 min	[59]
Electro-spinning method	1D nanofibers	Pollutant: RhB Dye (10 ppm) Light source: 300 W xenon-illuminator % degradation: 100% after 120 min	[60]
Electro-spinning method	1D micro-ribbons	Pollutant: RhB Dye (20 ppm) Light source: 500 W xenon lamp % degradation: 93.3% after 300 min	[61]
Solvothermal method through adjusting the solution pH	Red blood cell, flower-like microsphere, and dendrite morphologies	Pollutant: MB Dye (10 ppm) Light source: 500 W xenon lamp % degradation: dendrite-like < flower-like microsphere < red-blood-cell-like morphology after 180 min	[62]
Coprecipitation (500 °C for 5 h)	Cuboids	Pollutant: IC Dye (50 mL) Light source: fluorescence light 18 W % degradation: ~90% after 300 min	[63]
Precipitation (450 °C for 15 min)	Polyhedral	Pollutant: TBC (5 ppm) Light source: two visible lamps (15 W) % degradation: 97% after 300 min	[64]
Sol–gel method	Spherical structures	Pollutant: MO (15 ppm) Light source: 250 W halogen lamp % degradation: 98% after 50 min	[65]
Pechini sol–gel method	Rectangular cube-like, plate-like microstructures, plate-like nanostructures, nanorods, quasi-spherical structures	Pollutant: Thiophen (800 ppm) Light source: 400 W Osram lamp % degradation: 92% after 150 min	[66]
Modified one-step sol–gel method	Spherical	Pollutant: AB-113 (40 ppm) Light source: 1.6 kW xenon arc ozone-free lamp % degradation: (~99%) after 120 min	[67]

**Table 3 nanomaterials-13-01528-t003:** BiVO_4_-based semiconductor nanomaterials for VOCs degradation.

S. No.	Materials	VOCs Degraded	VOCs Concentration	Degradation of VOCs (%)	Degradation Time of VOCs (h)	Ref.
1	BiVO_4_	Isopropanol	160 ppm	88	12	[122]
2	BiVO_4_/TiO_2_	Benzene	260 ppm	84	8	[123]
3	BiVO_4_/CuO	Toluene	75 μL/L	85	6	[124]
4	V_2_O_5_/BiVO_4_/TiO_2_	Toluene	120 ppm	91	6	[125]
5	Quantum-sized BiVO_4_/TiO_2_ microflower	Toluene	-	89	6	[126]
6	BiVO_4_/RGO/Bi_2_O_3_	Toluene	25 ppm	95.6	6	[127]
7	BiVO_4_/P25	Ethylene	0.15 ppm	11.2	6	[128]
8	β-Bi_2_O_3_/BiVO_4_	o-DCB	-	70	6	[129]
9	Coral-like Z-scheme BiVO_4_/g-C_3_N_4_	Toluene	25 ppm	68.2	8	[130]
10	BiVO_4_/α-Fe_2_O_3_	Benzene	100 ppm	66.87	3.5	[131]
11	BiVO_4_/TiO_2_	Benzene	260 ppm	41	8	[132]
12	BiVO_4_ quantum tubes/rGO	HCHO	50 ppm	60	15 min	[133]
13	g-C_3_N_4_/BiVO_4_/tourmaline powder	Automobile exhaust gas (HC, NO, CO)	300–400 ppm (HC), 2.5–4% (NO), and 45–65 ppm (CO)	6.9 (HC) 7.2 (NO) 46.7 (CO)	1	[134]
14	Oxygen vacancies (OVs) introduced BiVO_4_/WO_3_/TiO_2_ nanotubes	Toluene	100 ppm	100	1	[135]
15	BiVO_4_ quantum dots/ZnO nanorod	HCHO	50 ppm	100	1	[136]
16	Ag/BiVO_4_	HCHO	10 ppm	84	5	[137]
17	ZnIn_2_S_4_-NiO/BiVO_4_	HCHO	1.5 mol/L	17 mmol/h	3	[138]
18	semi-coke activated carbon/ BiVO_4_–BiPO_4_–g-C_3_N_4_	Toluene	200 ppm	85.6	2	[139]
19	CdS-Au-{010}BiVO_4_{110}-MnOx	Toluene	4723 mg/m^3^	97.2	100 min	[140]

## Data Availability

Not applicable.

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
