# Peer review of "BiVO4 As a Sustainable and Emerging Photocatalyst: Synthesis Methodologies, Engineering Properties, and Its Volatile Organic Compounds Degradation Efficiency"

_nanomaterials, 2023, doi:10.3390/nano13091528_

Round 1

Reviewer 1 Report

This manuscript attempts to review the recent development in BiVO4 based materials and their potential visible light photocatalytic applications in degradation of VOCs. In general, the manuscript is suitable to the journal, but several issues should be addressed before it can be accepted for publication. Detailed comments are listed below:

1. The introduction should focus more on BiVO4 and other irrelevant discussion should be streamlined.

2. Incorrect formats appear throughout the article, for example, incorrect subscript, inconsistent letter fonts and table format. In addition, the English writing should be carefully polished.

3. Bandgap energy, VB position and CB position of bismuth-based semiconducting materials

should be added in Table 1 for comparison.

4. The degradation effects of BiVO4 with different morphologies on various pollutants should be supplemented in Table 2. In addition, it would be more clear if the information shown in Table 2 is presented in a Figure.

5. For fourth part, it is suggested to divide the paragraphs based on synthesis methods of BiVO4 photocatalyst. In addition, the review paper should not just list out what has been done before, but also need to give a summary based on certain logics.

6. Comparison between doping system and mixed phase systems should be added in Part 5. For this purpose, summary figures or tables are always necessary.

7. The degradation time of VOCs and the toxicity of their product should be taken into account in Table 3.

9. Perspective based on the state-of-art progress should be put forward at the end of the review.

10. Some related papers (i.e., Applied Catalysis B: Environmental 310 (2022) 121330; Rare Metals 41(2022) 3795-3802; Rare Metals 41 (2022) 3795-3802; Acta Physico-Chimica Sinica 37 (2021) 2011039; Journal of Hazardous Materials 384 (2020) 121494) should be cited and discussed.

Author Response

response to review is attached herewith.

Reviewer 2 Report

The future prospects of this material need to be elucidated.

Please discuss in detail the S-scheme and Z-scheme related to BiVO4.

Please point out more limitations of BiVO4 as a single catalyst and its importance in binary and ternary composite-based photocatalysts.

Please separate the synthesis part and photocatalytic performance for readers to understand better.

A comparison table is needed to understand the efficiency of photocatalysts in terms of quantum yield.

After the introductory part of the article by S V Prabhakar Vattikuti, the authors need to emphasize and put more emphasis on the importance of bismuth based photocatalysts.

Please stick to recent articles for reference articles, as several have already been reported.

Please reorganize the review article with the latest article.

Reviewer 3 Report

The paper may be accepted for publication after minor revision, taking into account:

- Fig. 11 and 12 should be clearer

- Please, check lines:  107, 174, 177-179, 180-181, 479

Round 2

Reviewer 1 Report

The authors have addressed the issues raised by me and could be accepted. 

Author Response

Thanks to the reviewer for recommending the manuscript for publication in the journal.